# SPATIO-TEMPORAL GRAPH SCATTERING TRANSFORM

**Chao Pan**
University of Illinois at Urbana-Champaign
Champaign, IL, USA
*chaopan2@illinois.edu*

**Siheng Chen** *
Shanghai Jiao Tong University
Shanghai, China
`sihengc@sjtu.edu.cn`

**Antonio Ortega**
University of Southern California
Los Angeles, CA, USA
*antonio.ortega@ee.usc.edu*

## ABSTRACT

Although spatio-temporal graph neural networks have achieved great empirical success in handling multiple correlated time series, they may be impractical in some real-world scenarios due to a lack of sufficient high-quality training data. Furthermore, spatio-temporal graph neural networks lack theoretical interpretation. To address these issues, we put forth a novel mathematically designed framework to analyze spatio-temporal data. Our proposed spatio-temporal graph scattering transform (ST-GST) extends traditional scattering transforms to the spatio-temporal domain. It performs iterative applications of spatio-temporal graph wavelets and nonlinear activation functions, which can be viewed as a forward pass of spatio-temporal graph convolutional networks without training. Since all the filter coefficients in ST-GST are mathematically designed, it is promising for the real-world scenarios with limited training data, and also allows for a theoretical analysis, which shows that the proposed ST-GST is stable to small perturbations of input signals and structures. Finally, our experiments show that i) ST-GST outperforms spatio-temporal graph convolutional networks by an increase of 35% in accuracy for MSR Action3D dataset; ii) it is better and computationally more efficient to design the transform based on separable spatio-temporal graphs than the joint ones; and iii) the nonlinearity in ST-GST is critical to empirical performance.

## 1 INTRODUCTION

Processing and learning from spatio-temporal data have received increasing attention recently. Examples include: i) skeleton-based human action recognition based on a sequence of human poses (Liu et al. (2019)), which is critical to human behavior understanding (Borges et al. (2013)), and ii) multi-agent trajectory prediction (Hu et al. (2020)), which is critical to robotics and autonomous driving (Shalev-Shwartz et al. (2016)). A common pattern across these applications is that data evolves in both spatial and temporal domains. This paper aims to analyze this type of data by developing novel spatio-temporal graph-based *data modeling* and *operations*.

**Spatio-temporal graph-based data modeling.** Graphs are often used to model data where irregularly spaced samples are observed over time. Good spatio-temporal graphs can provide informative priors that reflect the internal relationships within data. For example, in skeleton-based human action recognition, we can model a sequence of 3D joint locations as data supported on skeleton graphs across time, which reflects both the human physical constraints and temporal consistency (Yan et al. (2018)). Recent studies on modeling spatio-temporal graphs have followed either joint or separable processing frameworks. *Joint processing* is based on constructing a single spatio-temporal graph and processing (e.g., filtering) via operations on this graph (Kao et al. (2019); Liu et al. (2020)). In contrast, a *separable processing* approach works separately, and possibly with different operators, across the space and time dimension. In this case, independent graphs are used for space and

---

*This work was mainly done while Chao Pan and Siheng Chen were working at Mitsubishi Electric Research Laboratories (MERL).

time (Yan et al. (2018); Cheng et al. (2020)). However, no previous work thoroughly analyzes and compares these two constructions. In this work, we mathematically study these two types of graphs and justify the benefits of separable processing from both theoretical and empirical aspects.

**Spatio-temporal graph-based operations.** Graph operations can be performed once the graph structure is given. Some commonly used graph operations include the graph Fourier transform (Shuman et al. (2013)), and graph wavelets (Hammond et al. (2011)). It is possible to extend those operations to the spatio-temporal graph domain. For example, Grassi et al. (2017) developed the short time-vertex Fourier transform and spectrum-based time-vertex wavelet transform. However, those mathematically designed, linear operations show some limitations in terms of empirical performances. In comparison, many recent deep neural networks adopt trainable graph convolution operations to analyze spatio-temporal data (Yan et al. (2018); Liu et al. (2020)). However, most networks are designed through trial and error. It is thus hard to explain the rationale behind empirical success and further improve the designs (Monga et al. (2019)). In this work, to fill in the gap between mathematically designed linear transforms and trainable spatio temporal graph neural networks, we propose a novel spatio-temporal graph scattering transform (ST-GST), which is a mathematically designed, nonlinear operation.

Specifically, to characterize the spatial and temporal dependencies, we present two types of graphs corresponding to joint and separable designs. We then construct spatio-temporal graph wavelets based on each of these types of graphs. We next propose the framework of ST-GST, which adopts spatio-temporal graph wavelets followed by a nonlinear activation function as a single scattering layer. All the filter coefficients in ST-GST are mathematically designed beforehand and no training is required. We further show that i) a design based on separable spatio-temporal graph is more flexible and computationally efficient than a joint design; and ii) ST-GST is stable to small perturbations on both input spatio-temporal graph signals and structures. Finally, our experiments on skeleton-based human action recognition show that the proposed ST-GST outperforms spatio-temporal graph convolutional networks by 35% accuracy in MSR Action3D dataset.

We summarize the main contributions of this work as follows:

• We propose wavelets for both separable and joint spatio-temporal graphs. We show that it is more flexible and computationally efficient to design wavelets based on separable spatio-temporal graphs;

• We propose a novel spatio-temporal graph scattering transform (ST-GST), which is a non-trainable counterpart of spatio-temporal graph convolutional networks and a nonlinear version of spatio-temporal graph wavelets. We also theoretically show that ST-GST is robust and stable in the presence of small perturbations on both input spatio-temporal graph signals and structures;

• For skeleton-based human action recognition, our experiments show that: i) ST-GST can achieve similar or better performances than spatio-temporal graph convolutional networks and other non-deep-learning approaches in small-scale datasets; ii) separable spatio-temporal scattering works significantly better than joint spatio-temporal scattering; and iii) ST-GST significantly outperforms spatio-temporal graph wavelets because of the nonlinear activation function.

## 2 RELATED WORK

**Scattering transforms.** Convolutional neural networks (CNNs) use nonlinearities coupled with trained filter coefficients and are well known to be hard to analyze theoretically (Anthony & Bartlett (2009)). As an alternative, Mallat (2012); Bruna & Mallat (2013) propose scattering transforms, which are non-trainable versions of CNNs. Under admissible conditions, the resulting transform enjoys both great performance in image classification and appealing theoretical properties. These ideas have been extended to the graph domain (Gama et al. (2019a); Zou & Lerman (2020); Gao et al. (2019); Ioannidis et al. (2020)). Specifically, the graph scattering transform (GST) proposed in (Gama et al. (2019a)) iteratively applies predefined graph filter banks and element-wise nonlinear activation function. In this work, we extend classical scattering transform to the spatio-temporal domain and provide a new mathematically designed transform to handle spatio-temporal data. The key difference between GST and our proposed ST-GST lies in the graph filter bank design, where ST-GST needs to handle both spatial and temporal domains.

**Spatio-temporal neural networks.** Deep neural networks have been adapted to operate on spatio-temporal domain. For example, Liu et al. (2019) uses LSTM to process time series information, while ST-GCN (Yan et al. (2018)) combines a graph convolution layer and a temporal convolution

layer as a unit computational block in the network architecture. However, those networks all require a huge amount of high-quality labeled data, and training them is computationally expensive, which may make them impractical for many real-world scenarios. Furthermore, many architectures are designed through trial and error, making it hard to justify the design choices and further improve them. In this work, the proposed ST-GST is a nonlinear transform with a forward procedure similar to that of ST-GCN. However, ST-GST does not require any training, which is useful in many applications where only limited training data is available. Furthermore, since all filter coefficients in ST-GST are predefined, it allows us to perform theoretical analysis and the related conclusions potentially shed some light on the design of spatio-temporal networks.

**Skeleton-based human action recognition.** Conventional skeleton-based action recognition models learn semantics based on hand-crafted features (Wang et al. (2012)). To handle time series information, some recurrent-neural-network-based models are proposed to capture the temporal dependencies between consecutive frames (Kim & Reiter (2017)). Recently, graph-based approaches have gained in popularity while achieving excellent performance (Yan et al., 2018; Li et al., 2019). In this work, our experiments focus on this task and show that ST-GST outperforms the state-of-the-art spatio-temporal graph neural networks, like MS-G3D (Liu et al., 2020), on small-scale datasets.

## 3 SPATIO-TEMPORAL GRAPH SCATTERING TRANSFORM

In this section, we first define spatio-temporal graph structures and signals. We next design our spatio-temporal graph wavelets. Finally, we present ST-GST.

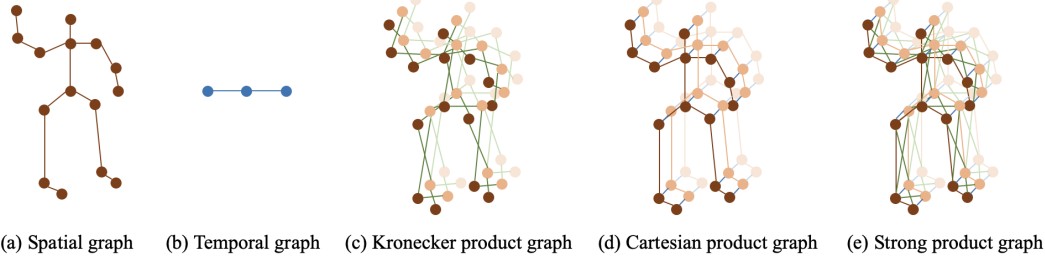

(a) Spatial graph    (b) Temporal graph    (c) Kronecker product graph    (d) Cartesian product graph    (e) Strong product graph

Figure 1: Visualization of spatial graph, temporal graph and three commonly used product graphs.

### 3.1 SPATIO-TEMPORAL GRAPH STRUCTURES AND SIGNALS

Spatio-temporal data can be represented as a matrix $\mathbf{X} \in \mathbb{R}^{N \times T}$, where $N$ is the number of the spatial positions and $T$ is the number of time stamps. In this matrix, each row is a time-series for a spatial node, and each column is a spatial signal at a certain time stamp. Note that the index of spatial information can be arbitrary: we will associate each spatial location to a vertex on the spatial graph, and the edges will provide information about the relative position of the nodes. We can reshape the matrix to form a vector $\mathbf{x}$ of length $NT$, where the element $\mathbf{x}_{(s,t)} := \mathbf{x}_{(s-1)T+t}$ is the feature value corresponding to the $s$-th vertex at time $t$. To construct a spatio-temporal graph, we create connections based on physical constraints. For example, for skeleton-based action recognition, the spatial graph is the human skeleton graph, reflecting bone connections; see Fig. 1(a); and the temporal graph is a line graph connecting consecutive time stamps; see Fig. 1(b).

As a starting point, we choose a spatial graph $\mathcal{G}_s = (\mathcal{V}_s, \mathcal{E}_s, \mathbf{A}_s)$ with $|\mathcal{V}_s| = N$, reflecting the graph structure of each column in $\mathbf{X}$ and a temporal graph $\mathcal{G}_t = (\mathcal{V}_t, \mathcal{E}_t, \mathbf{A}_t)$ with $|\mathcal{V}_t| = T$, reflecting the graph structure of each row in $\mathbf{X}$. The *separable spatio-temporal design* is achieved by processing the columns and rows of $\mathbf{X}$ separately based on their respective graphs.

As an alternative, a product graph, denoted as $\mathcal{G} = \mathcal{G}_s \diamond \mathcal{G}_t = (\mathcal{V}, \mathcal{E}, \mathbf{A})$ can be constructed to unify the relations in both the spatial and temporal domains, allowing us to process data *jointly* across space and time. The product graph $\mathcal{G}_s \diamond \mathcal{G}_t$ has $|\mathcal{V}| = NT$ nodes and an appropriately defined $NT \times NT$ adjacency matrix $\mathbf{A}$. The operation $\diamond$ interweaves two graphs to form a unifying graph structure. The edge weight $\mathbf{A}_{(s_1,t_1),(s_2,t_2)} := \mathbf{A}_{(s_1-1)T+t_1,(s_2-1)T+t_2}$ characterizes the relation, such as similarity or dependency, between the $s_1$-th spatial node at the $t_1$-th time stamp and the $s_2$-th spatial node at the $t_2$-th time stamp. There are three commonly used product graphs (Sandryhaila & Moura, 2014): i) Kronecker product: $\mathcal{G} = \mathcal{G}_s \otimes \mathcal{G}_t$ with graph adjacency matrix as $\mathbf{A} = \mathbf{A}_s \otimes \mathbf{A}_t$ and

$\otimes$ represents the Kronecker product of matrices; see Fig 1(c); ii) Cartesian product: $\mathcal{G} = \mathcal{G}_s \times \mathcal{G}_t$ with $\mathbf{A} = \mathbf{A}_s \otimes \mathbf{I}_T + \mathbf{I}_N \otimes \mathbf{A}_t$; see Fig 1(d); and iii) strong product: $\mathcal{G} = \mathcal{G}_s \boxtimes \mathcal{G}_t$ with $\mathbf{A} = \mathbf{A}_s \otimes \mathbf{A}_t + \mathbf{A}_s \otimes \mathbf{I}_T + \mathbf{I}_N \otimes \mathbf{A}_t$, which can be viewed as a combination of Kronecker and Cartesian products; see Fig 1(e). The *joint spatio-temporal design* is achieved based on a product graph.

In this paper, we consider designs based on both separable graphs and product graphs.

## 3.2  SPATIO-TEMPORAL GRAPH FILTERING

We now show two graph filter designs, separable and joint filtering, based on the corresponding spatio-temporal graphs we just described. For each design, we first define the spatio-temporal graph shift, which is the most elementary graph filter and defines how information should propagate in a spatio-temporal graph. We then propose spatio-temporal graph filtering in both graph vertex and graph spectral domains.

**Separable graph filters.** Given the spatial graph $\mathcal{G}_s = (\mathcal{V}_s, \mathcal{E}_s, \mathbf{A}_s)$ and the temporal graph $\mathcal{G}_t = (\mathcal{V}_t, \mathcal{E}_t, \mathbf{A}_t)$, let the spatial graph shift be $\mathbf{S}_s = \mathbf{A}_s$ and the temporal graph shift be $\mathbf{S}_t = \mathbf{A}_t$ [1]. For simplicity, we focus on the symmetric graph shifts. For a spatio-temporal graph signal, spatial and temporal graph filtering work as, $\mathbf{H}(\mathbf{S}_s)\mathbf{X} = \sum_{p=0}^{P-1} h_p \mathbf{S}_s^p \mathbf{X}$ and $\mathbf{X}\mathbf{G}^\top(\mathbf{S}_t) = \mathbf{X}(\sum_{q=0}^{Q-1} g_q \mathbf{S}_t^q)^\top$, where $h_p$ are $g_q$ are spatial and temporal filter coefficients, respectively. In each modality, the graph filter is a polynomial of the graph shift. The polynomial orders $P$ and $Q$ control the length of filters in the spatial and temporal modalities, respectively. Note that these two values can be chosen to be different, which provides additional design flexibility. Then, a separable spatio-temporal graph filtering operation can be defined as

$$\mathbf{H}(\mathbf{S}_s)\mathbf{X}\mathbf{G}^\top(\mathbf{S}_t) := \left(\sum_{p=0}^{P-1} h_p \mathbf{S}_s^p\right)\mathbf{X}\left(\sum_{q=0}^{Q-1} g_q \mathbf{S}_t^q\right)^\top = (\mathbf{H}(\mathbf{S}_s) \otimes \mathbf{G}(\mathbf{S}_t))\,\mathbf{x}, \tag{1}$$

where the second equality follows from the property: $\mathbf{M}_1 \mathbf{X} \mathbf{M}_2^T = (\mathbf{M}_1 \otimes \mathbf{M}_2)\mathbf{x}$.

We can also represent the filtering process in the graph spectral domain. Let the eigen-decomposition of the spatial and temporal graphs be $\mathbf{S}_s = \mathbf{V}_s \mathbf{\Lambda}_s \mathbf{V}_s^\top$ and $\mathbf{S}_t = \mathbf{V}_t \mathbf{\Lambda}_t \mathbf{V}_t^\top$, respectively, where $\mathbf{V}_s \in \mathbb{R}^{N \times N}, \mathbf{V}_t \in \mathbb{R}^{T \times T}$ form the spatial and temporal graph Fourier bases. The elements along the diagonals of $\mathbf{\Lambda}_s, \mathbf{\Lambda}_t$ represent the spatial and temporal graph frequencies. We have $\mathbf{H}(\mathbf{S}_s)\mathbf{X} = \mathbf{V}_s \sum_{p=0}^{P-1} h_p \mathbf{\Lambda}_s^p \mathbf{V}_s^\top \mathbf{X} = \mathbf{V}_s \mathbf{H}(\mathbf{\Lambda}_s) \mathbf{V}_s^\top \mathbf{X}$, $\mathbf{X}\mathbf{G}^\top(\mathbf{S}_t) = \mathbf{X} \mathbf{V}_t \sum_{q=0}^{Q-1} g_q \mathbf{\Lambda}_t^q \mathbf{V}_t^\top = \mathbf{X}\mathbf{V}_t \mathbf{G}(\mathbf{\Lambda}_t)\mathbf{V}_t^\top$. Letting $\mathbf{V} = \mathbf{V}_s \otimes \mathbf{V}_t$, the spectral representation of the separable spatio-temporal graph filtering is then $(\mathbf{V}_s \mathbf{H}(\mathbf{\Lambda}_s)\mathbf{V}_s^\top)\mathbf{X}(\mathbf{V}_t \mathbf{G}(\mathbf{\Lambda}_t)\mathbf{V}_t^\top)^\top = \mathbf{V}(\mathbf{H}(\mathbf{\Lambda}_s) \otimes \mathbf{G}(\mathbf{\Lambda}_t))\mathbf{V}^\top \mathbf{x}$.

**Joint graph filters.** Given the joint graph structure $\mathcal{G} = \mathcal{G}_s \diamond \mathcal{G}_t = (\mathcal{V}, \mathcal{E}, \mathbf{A})$, let the spatio-temporal graph shift be $\mathbf{S} = \mathbf{A}$. Then, a joint spatio-temporal graph filtering operation can be defined as:

$$\mathbf{H}(\mathbf{S})\mathbf{x} = \sum_{k=0}^{K-1} h_k \mathbf{S}^k \mathbf{x} = \mathbf{V}\left(\sum_{k=0}^{K-1} h_k \mathbf{\Lambda}^k\right)\mathbf{V}^\top \mathbf{x} = \mathbf{V}\mathbf{H}(\mathbf{\Lambda})\mathbf{V}^\top \mathbf{x}, \tag{2}$$

where $h_k$ is the filter coefficient. The kernel function $h(\lambda) = \sum_{k=0}^{K-1} h_k \lambda^k$ is applied to each diagonal element of $\mathbf{\Lambda}$ to obtain $\mathbf{H}(\mathbf{\Lambda})$. Here $h(\lambda)$ is independent of any specific graph structure, and characterizes the filter response in the graph spectral domain. Note that $\mathbf{h} = (h_0, \cdots, h_{K-1})$, $h(\lambda)$ and $\mathbf{H}(\cdot)$ are essentially the same thing, and are used interchangeably in this paper.

It is worth pointing out that these three product graphs share the same form of joint spatio-temporal graph filtering (2) as well as the same graph Fourier bases $\mathbf{V} = \mathbf{V}_s \otimes \mathbf{V}_t$. Following from (2), the spectral representation of the joint spatio-temporal graph filtering can be formulated as

$$\text{Kronecker product:}\quad \mathbf{H}(\mathbf{S}) = \mathbf{V}(\mathbf{H}(\mathbf{\Lambda}_s \otimes \mathbf{\Lambda}_t))\mathbf{V}^\top,$$

$$\text{Cartesian product:}\quad \mathbf{H}(\mathbf{S}) = \mathbf{V}(\mathbf{H}(\mathbf{\Lambda}_s \otimes \mathbf{I}_T + \mathbf{I}_N \otimes \mathbf{\Lambda}_t))\mathbf{V}^\top,$$

$$\text{Strong product:}\quad \mathbf{H}(\mathbf{S}) = \mathbf{V}(\mathbf{H}(\mathbf{\Lambda}_s \otimes \mathbf{\Lambda}_t + \mathbf{\Lambda}_s \otimes \mathbf{I}_T + \mathbf{I}_N \otimes \mathbf{\Lambda}_t))\mathbf{V}^\top.$$

---

[1] Some other choices of a graph shift include the graph Laplacian matrix, graph transition matrix and their normalized counterparts. Adjacency matrix is considered here for notation simplicity.

**Comparison between separable and joint graph filters.** First of all, we stress that both separable and joint spatio-temporal graph filtering share the same Fourier bases, meaning that they share the same frequency space and their difference only comes from the frequency responses.

Second, designing filters based on separable spatio-temporal graphs provides additional flexibility. Although it is conceptually simple to design graph filters directly on product graphs, the eigenvalues along the spatial and temporal domains are tied together, making it difficult to adjust the frequency responses independently for the two modalities. Moreover, two domains are forced to share the same set of filter coefficients and length. Take a filter defined on Kronecker product graph as an example. By expanding the term $\mathbf{H}(\mathbf{\Lambda}_s \otimes \mathbf{\Lambda}_t)$ we can have that $\mathbf{H}(\mathbf{\Lambda}_s \otimes \mathbf{\Lambda}_t) = \sum_{k=0}^{K-1} h_k(\mathbf{\Lambda}_s \otimes \mathbf{\Lambda}_t)^k = \sum_{k=0}^{K-1} h_k(\mathbf{\Lambda}_s^k \otimes \mathbf{\Lambda}_t^k)$. This shows that the filter coefficients are applied on the product of spatial and temporal eigenvalues, making it hard to decompose and interpret the functionality of the filter in single modality. Such limitations make them less practical for spatio-temporal signals which might have distinct patterns in each of the two modalities. This problem is overcome by separable graph filtering, where different filters are applied to each modality. The flexibility of separable graph filters means that one can design different filters ($\mathbf{h}$ and $\mathbf{g}$) with independent filter lengths ($P$ and $Q$) in the spatial and temporal domains. However, it is worth pointing out that the representation power of these two formulations does not have a clear relationship that one is a subset of the other. Consider a joint graph filter designed on a strong product graph with length $K = 3$. The filter kernel is defined as $\mathbf{H}(\mathbf{\Lambda}_s \otimes \mathbf{\Lambda}_t + \mathbf{\Lambda}_s \otimes \mathbf{I}_T + \mathbf{I}_N \otimes \mathbf{\Lambda}_t) = \sum_{k=0}^{2} h_k(\mathbf{\Lambda}_s \otimes \mathbf{\Lambda}_t + \mathbf{\Lambda}_s \otimes \mathbf{I}_T + \mathbf{I}_N \otimes \mathbf{\Lambda}_t)^k$. Similarly, the kernel of a separable graph filter with $P = Q = 3$ can be written as $\mathbf{H}(\mathbf{\Lambda}_s) \otimes \mathbf{G}(\mathbf{\Lambda}_t) = (\sum_{p=0}^{2} h_p\mathbf{\Lambda}_s^p) \otimes (\sum_{q=0}^{2} g_q\mathbf{\Lambda}_t^q)$. By expanding the expression and rearranging the coefficients, one can obtain the coefficient matrices for the joint graph filter and the separable graph filter, $C_1$ and $C_2$, respectively; that is,

$$C_1 = \begin{bmatrix} h_0 & h_1 & h_2 \\ h_1 & h_1 + 2h_2 & 2h_2 \\ h_2 & 2h_2 & h_2 \end{bmatrix}, \quad C_2 = \begin{bmatrix} h_0g_0 & h_0g_1 & h_0g_2 \\ h_1g_0 & h_1g_1 & h_1g_2 \\ h_2g_0 & h_2g_1 & h_2g_2 \end{bmatrix} = \begin{bmatrix} h_0 \\ h_1 \\ h_2 \end{bmatrix} \begin{bmatrix} g_0 & g_1 & g_2 \end{bmatrix},$$

where $(i, j)$-th element means the coefficient of term $\mathbf{\Lambda}_s^{i-1} \otimes \mathbf{\Lambda}_t^{j-1}$.

On one hand, it is obvious that $C_2$ is always a rank 1 matrix, while $C_1$ could have rank 1, 2, or 3. So $C_1$ is not a special case of $C_2$. On the other hand, $C_1$ is always a symmetric matrix, but $C_2$ can be either symmetric or non-symmetric, depending on the choices of $\mathbf{h}$ and $\mathbf{g}$. So $C_2$ is also not a special case of $C_1$. Therefore, the families spanned by two designs do not have any simple relationship that one is a subset of the other. Similar conclusions hold for the Kronecker and Cartesian products.

Third, designing based on separable spatio-temporal graphs is computationally more efficient. In a separable graph filtering process, we only need to deal with two small matrix multiplications (1), instead of one large matrix-vector multiplication (2), reducing the computational cost from $O(N^2T^2)$ to $O(NT(N + T))$.

In short, the joint and separable graph filters are two different design methods for spatio-temporal graphs. Though the representation power of separable graph filters is not necessarily much stronger than joint ones, separable design enjoys the flexibility, computation efficiency and straightforward interpretation. Empirical performances also show that the separable design outperforms the joint one; see Section 5. Note that this separable design coincides with the basic module widely used in spatio-temporal graph convolutional networks Li et al. (2019), which consists of one graph convolution layer followed by one temporal convolution layer.

### 3.3 SPATIO-TEMPORAL GRAPH WAVELETS

In time-series analysis and image processing, wavelets are one of the best tools to design filter banks, allowing us to trade-off between the time-frequency resolutions and touching the lower bound of uncertainty principle of the time-frequency representations (Akansu & Haddad, 2000). Inspired by this, we propose spatio-temporal graph wavelets, which include a series of mathematically designed graph filters to provide multiresolution analysis for spatio-temporal graph signals. The proposed spatio-temporal graph wavelets are later used at each layer in the proposed ST-GST framework. Based on two types of graph structures, we consider two designs: separable and joint wavelets.

**Separable graph wavelets.** Based on separable spatio-temporal graph filtering (1), we are able to design spatial graph wavelets, $\{\mathbf{H}_{j_1}(\mathbf{S}_s) = \sum_{p=0}^{P-1} h_p^{(j_1)}\mathbf{S}_s^p\}_{j_1=1}^{J_s}$, and temporal graph wavelets,

$\{\mathbf{G}_{j_2}(\mathbf{S}_t) = \sum_{q=0}^{Q-1} g_q^{(j_2)} \mathbf{S}_t^q\}_{j_2=1}^{J_t}$, separately. For each modality, the filter at scale $j$ is defined as $\mathbf{H}_j(\mathbf{S}) = \mathbf{S}^{2^{j-1}} - \mathbf{S}^{2^j} = \mathbf{S}^{2^{j-1}}(\mathbf{I} - \mathbf{S}^{2^{j-1}})$. There are also many other off-the-shelf graph wavelets we can choose from. More discussion about wavelets and their properties can be found in Appendix A. Since two modalities are designed individually, the number of wavelet scales for each modality could be different. This is important in practice because the number of time samples $T$ is normally larger than the number of spatial nodes $N$. For each node in spatio-temporal graph, using different wavelet scales in the two domains allows for more flexibility to diffuse the signal with its neighbors. Based on this construction, when we choose $J_s$ and $J_t$ scales for spatial and temporal domains, respectively, the overall number of scales for spatio-temporal wavelets is then $J = J_s \times J_t$.

**Joint graph wavelets.** When the joint filtering (2) is chosen, we can directly apply existing graph wavelet designs, such as the spectral graph wavelet transform (Hammond et al., 2011).

### 3.4 SPATIO-TEMPORAL GRAPH SCATTERING TRANSFORM

The proposed ST-GST is a nonlinear version of spatio-temporal graph wavelets, which iteratively uses wavelets followed by a nonlinearity activation function. ST-GST includes three components: (i) spatio-temporal graph wavelets, (ii) a pointwise nonlinearity activation function $\sigma(\cdot)$, and (iii) a low-pass pooling operator $U$. These operations are performed sequentially to extract representative features from input spatio-temporal graph signal $\mathbf{X}$. The main difference between ST-GST and spatio-temporal graph wavelets is the application of nonlinear activation at each layer. The nonlinear transformation disperses signals through the graph spectrum, producing more patterns in spectrum.

**Separable ST-GST.** Let $\mathbf{Z} \in \mathbb{R}^{N \times T}$ be a spatio-temporal graph signal. At each scattering layer, we sequentially use spatial graph wavelets $\{\mathbf{H}_{j_1}\}_{j_1=1}^{J_s}$ and temporal wavelets $\{\mathbf{G}_{j_2}\}_{j_2=1}^{J_t}$ to convolve with $\mathbf{Z}$. Since each graph filter generates a new spatio-temporal graph signal, separable spatio-temporal graph filtering generates $J = J_s \times J_t$ spatio-temporal graph signals. Then, the nonlinear activation is applied for each spatio-temporal graph signal. For example, the $(j_1, j_2)$-th signal is $\mathbf{Z}_{(j_1,j_2)} = \sigma(\mathbf{H}_{j_1}(\mathbf{S}_s)\mathbf{Z}\mathbf{G}_{j_2}^\top(\mathbf{S}_t))$. We can treat each filtered spatio-temporal graph signal as one tree node. Given $\mathbf{Z}$ as the parent node, a scattering layer produces $J$ children nodes.

To construct ST-GST, we first initialize the input data $\mathbf{Z}_0 = \mathbf{X}$ be the root of the scattering tree; and then, we recursively apply scattering layers at each node to produce children nodes, growing a scattering tree; see Fig. 2. We can index all the nodes in this scattering tree by a unique path from the root to each node. For example, $p^{(\ell)} = ((j_1^{(1)}, j_2^{(1)}), \ldots, (j_1^{(\ell)}, j_2^{(\ell)}))$ is the path from root to one tree node in the $\ell$-th layer, and the signal associated with it is $\mathbf{Z}_{(p^{(\ell)})}$. Data matrix $\mathbf{Z}_{(p^{(\ell)})}$ is then summarized by an pooling operator $U(\cdot)$ to obtain a lower-dimensional vector $\phi_{(p^{(\ell)})} = U\left(\mathbf{Z}_{(p^{(\ell)})}\right)$. Various pooling methods can lead to different dimensions of scattering features. Common choices for $U(\cdot)$ include average in the spatial domain ($\mathbf{U} = \frac{1}{N}\mathbf{1}_{1\times N}, \phi = \mathbf{U}\mathbf{Z} \in \mathbb{R}^T$), average in the temporal domain ($\mathbf{U} = \frac{1}{T}\mathbf{1}_{T\times 1}, \phi = \mathbf{Z}\mathbf{U} \in \mathbb{R}^N$) and average in both modalities ($\mathbf{U} = \frac{1}{NT}\mathbf{1}_{N\times T}, \phi = \mathbf{U} \circ \mathbf{Z} \in \mathbb{R}$), where $\circ$ represent Hadamard product. Finally, all scattering features $\phi_{(p^{(\ell)})}$ are concatenated to construct a scattering feature map $\Phi(\mathbf{S}_s, \mathbf{S}_t, \mathbf{X}) := \{\{\phi_{(p^{(\ell)})}\}_{\text{all } p^{(\ell)}}\}_{\ell=0}^{L-1}$,

**Joint ST-GST.** Since we deal with a unifying graph, we can use the spatio-temporal product graph directly in combination with the ordinary graph scattering transform (Gama et al. (2019b)).

**Comparison with ST-GCNs.** One distinct difference between ST-GST and ST-GCNs lies in the fact that the trainable graph convolution in each layer of ST-GCN performs the multiplication between a spatial graph shift and the feature matrix, which only extracts low-frequency information over the graph; while ST-GST leverages multiple spatio-temporal graph filters to cover multiple frequency bands. Furthermore, predefined filter coefficients conform a frame (3) in each layer of ST-GST, which is crucial for establishing the stability of ST-GST as shown in next section.

## 4 THEORETICAL ANALYSIS

Stability is the key to designing robust and reliable algorithms. However, since the training process of ST-GCNs is data-driven, it might be vulnerable to small perturbations added to training data, which may lead to significant degradation in practice. Trainable parameters make it hard to develop a theoretical analysis for ST-GCNs. In contrast, here we show that the proposed separable ST-GST is stable to perturbations on both spatio-temporal graph signals and structures. All proofs of statements

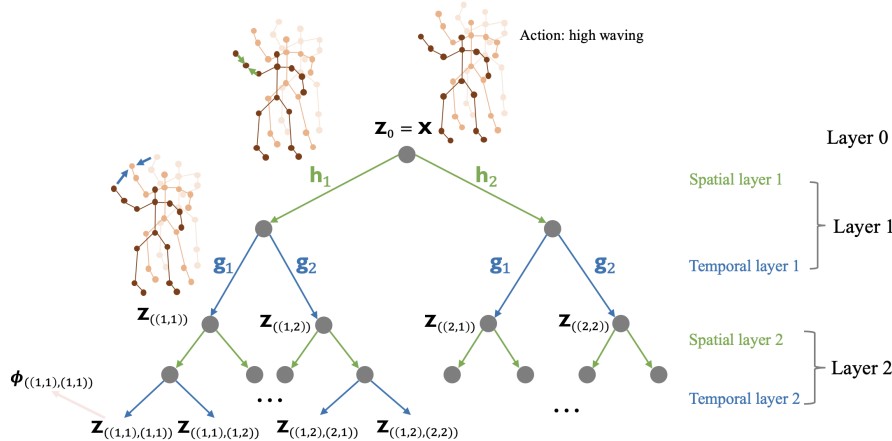

Figure 2: Scattering tree of separable ST-GST with $L = 3, J_s = J_t = 2$.

in this section are explained thoroughly in Appendix B. Unless specified, $\|\mathbf{x}\|$ is the $\ell_2$ norm for vector $\mathbf{x}$, while $\|\mathbf{X}\|$ and $\|\mathbf{X}\|_2$ are the Frobenius and spectral norm for matrix $\mathbf{X}$, respectively.

Here we show the results for separable spatio-temporal graph scattering transform. But all the results can be extended to the joint version. Before introducing perturbations, we first show that separable spatio-temporal graph wavelets also satisfy certain frame bounds. Thus, with a separable construction, we can control bound constants for spatio-temporal wavelet and build tight frames.

**Lemma 1.** Let $\{\mathbf{H}_{j_1}\}_{j_1=1}^{J_s}$ and $\{\mathbf{G}_{j_2}\}_{j_2=1}^{J_t}$ be the wavelet filter bank for spatial domain and for temporal domain, respectively. Both satisfy frame properties such that for any $\mathbf{x} \in \mathbb{R}^N$ and $\mathbf{y} \in \mathbb{R}^T$,

$$A_1^2\|\mathbf{x}\|^2 \leq \sum_{j_1=1}^{J_s} \|\mathbf{H}_{j_1}(\mathbf{S}_s)\mathbf{x}\|^2 \leq B_1^2\|\mathbf{x}\|^2, \quad A_2^2\|\mathbf{y}\|^2 \leq \sum_{j_2=1}^{J_t} \|\mathbf{G}_{j_2}(\mathbf{S}_t)\mathbf{y}\|^2 \leq B_2^2\|\mathbf{y}\|^2, \quad (3)$$

Then, for any $\mathbf{Z} \in \mathbb{R}^{N \times T}$ and its corresponding reshaped vector $\mathbf{z} \in \mathbb{R}^{NT}$, it holds that

$$A_1^2 A_2^2\|\mathbf{Z}\|^2 \leq \sum_{j_1,j_2=1}^{J_s,J_t} \|(\mathbf{H}_{j_1}(\mathbf{S}_s) \otimes \mathbf{G}_{j_2}(\mathbf{S}_t))\mathbf{z}\|^2 = \sum_{j_1,j_2=1}^{J_s,J_t} \left\|\mathbf{H}_{j_1}(\mathbf{S}_s)\mathbf{Z}\mathbf{G}_{j_2}^\top(\mathbf{S}_t)\right\|^2 \leq B_1^2 B_2^2\|\mathbf{Z}\|^2.$$

Lemma 1 guarantees that separable design can also lead to valid wavelets. Furthermore, when we choose both spatial $\{\mathbf{H}_{j_1}\}_{j_1=1}^{J_s}$ and temporal $\{\mathbf{G}_{j_2}\}_{j_2=1}^{J_t}$ to be tight frames with $A_1 = B_1 = A_2 = B_2 = 1$ (Shuman et al., 2015), the resulting separable wavelet also conforms a tight frame. In later context, denote $B = B_1 \times B_2$ as the frame bound constant for separable spatio-temporal graph wavelet, and separable ST-GST is configured with $L$ layers and $J = J_s \times J_t$ scales at each layer.

### 4.1 STABILITY TO PERTURBATION OF SPATIO-TEMPORAL GRAPH SIGNALS

Consider the perturbed spatio-temporal graph signal $\widetilde{\mathbf{X}} = \mathbf{X} + \mathbf{\Delta} \in \mathbb{R}^{N \times T}$, where $\mathbf{\Delta} \in \mathbb{R}^{N \times T}$ is the perturbation matrix. Such an additive model can represent measurement noise caused by devices or adversarial perturbations added manually. Theorem 1 shows that the feature map for perturbed signal will not deviate much from original feature map under small input perturbations.

**Theorem 1.** Consider the additive noise model for input data $\mathbf{X}$, it then holds that

$$\frac{\|\Phi(\mathbf{S}_s, \mathbf{S}_t, \mathbf{X}) - \Phi(\mathbf{S}_s, \mathbf{S}_t, \widetilde{\mathbf{X}})\|}{\sqrt{T \sum_{\ell=0}^{L-1} J^\ell}} \leq \frac{1}{\sqrt{NT}} \sqrt{\frac{\sum_{\ell=0}^{L-1} B^{2\ell}}{\sum_{\ell=0}^{L-1} J^\ell}} \|\mathbf{\Delta}\|. \quad (4)$$

The difference of output is normalized by the squared root of dimension of the final feature map. Note that we can construct spatio-temporal wavelet easily with $B = 1$ when spatial and temporal wavelets are both tight frames, then the normalized bound presented in (4) indicates that the transform is insensitive to perturbations on input signals as the factor is much smaller than 1.

|  | Method | Accuracy (%) |
|---|---|---|
| GNNs | GFT+TPM | 74.0 |
| | HDM | 81.8 |
| | Temporal Conv. | 72.1 |
| | ST-GCN (fixed topology) | 52.0 |
| | MS-G3D (learnable topology) | 82.2 |
| Scattering | Separable ST-GST (5, 10, 3) | 81.4 |
| | Separable ST-GST (5, 20, 3) | **87.0** |
| | Joint Kronecker ST-GST (15, 3) | 61.0 |
| | Joint Cartesian ST-GST (15, 3) | 59.1 |
| | Joint Strong ST-GST (15, 3) | 61.7 |

Table 1: Classification accuracy (MSR Action3D with 288 training and 269 testing samples).

|  | Method | Accuracy (%) |
|---|---|---|
| GNNs | Deep LSTM | 60.7 |
| | PA-LSTM | 62.9 |
| | ST-LSTM+TG | 69.2 |
| | Temporal Conv. | 74.3 |
| | ST-GCN (fixed topology) | **75.8** |
| Scattering | Separable ST-GST (5, 20, 2) | 68.7 |
| | Separable ST-GST (5, 20, 3) | 73.1 |
| | Joint Kronecker ST-GST (15, 3) | 55.7 |
| | Joint Cartesian ST-GST (15, 3) | 56.2 |
| | Joint Strong ST-GST (15, 3) | 57.1 |

Table 2: Classification accuracy (NTU-RGB+D with $40,320$ training and $16,560$ testing samples).

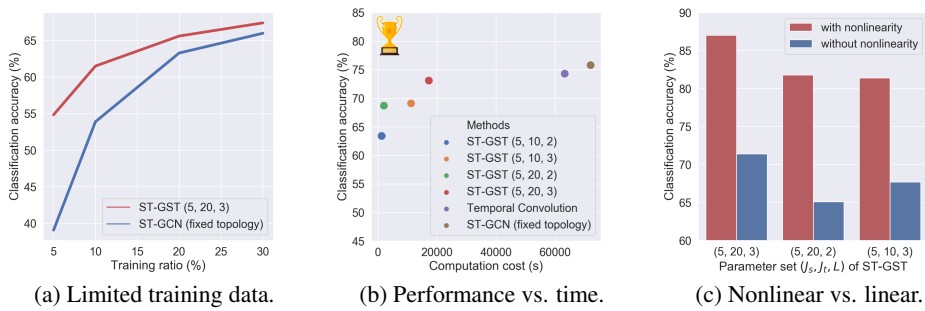

(a) Limited training data.  (b) Performance vs. time.  (c) Nonlinear vs. linear.

Figure 3: Performance comparisons with various settings about training ratio, time and nonlinearity.

## 4.2 STABILITY TO PERTURBATION OF SPATIO-TEMPORAL GRAPH STRUCTURES

Perturbations on the underlying graph usually happen when the graph is unknown or when the graph changes over time (Segarra et al., 2017). Since such kind of perturbations usually happen in spatial domain, here we simply consider the structure perturbations on the spatial graph only. Specifically, we consider the perturbed spatial graph $\widehat{\mathbf{S}}_s = \mathbf{S}_s + \mathbf{E}^\top \mathbf{S}_s + \mathbf{S}_s \mathbf{E}$, where $\mathbf{E}$ is the perturbation matrix and temporal graph $\mathbf{S}_t$ is not changed. Detailed descriptions see Appendix B.

**Theorem 2.** Suppose eigenvalues $\{m_i\}_{i=1}^N$ of $\mathbf{E} \in \mathbb{R}^{N \times N}$ are organized in order such that $|m_1| \leq |m_2| \leq \cdots \leq |m_N|$, satisfying $|m_N| \leq \epsilon/2$ and $|m_i/m_N - 1| \leq \epsilon$ for $\epsilon > 0$ and all $i$'s. Suppose spatial filter bank $\{\mathbf{H}_{j_1}\}_{j_1=1}^{J_s}$ satisfies $\max_i |\lambda h_i'(\lambda)| \leq C$ and temporal filter bank $\{\mathbf{G}_{j_2}\}_{j_2=1}^{J_t}$ satisfies limited spectral response $\max_i |g_i(\lambda)| \leq D$. It then holds that

$$\frac{\|\Phi(\mathbf{S}_s, \mathbf{S}_t, \mathbf{X}) - \Phi(\widehat{\mathbf{S}}_s, \mathbf{S}_t, \mathbf{X})\|}{\sqrt{T \sum_{\ell=0}^{L-1} J^\ell}} \leq \frac{\epsilon C D}{B\sqrt{NT}} \sqrt{\frac{\sum_{\ell=0}^{L-1} \ell^2 (B^2 J)^\ell}{\sum_{\ell=0}^{L-1} J^\ell}} \|\mathbf{X}\|, \tag{5}$$

where $\epsilon$ characterizes the perturbation level. Theorem 2 shows that ST-GST is a stable transform also for structure deformation, as the norm of change of feature map is linear in $\epsilon$. It is worth pointing out that the upper bound in both Theorems 1 and 2 only depend on the choice of filter banks and structure of scattering tree, instead of quantities related with specific graph support $\mathbf{S}_s$ and $\mathbf{S}_t$ that are shown in previous works (Gama et al., 2019b; Zou & Lerman, 2020; Levie et al., 2019).

## 5 EXPERIMENTAL RESULTS

We now evaluate the performance of proposed ST-GST in skeleton-based action recognition task.

**Experimental setup.** The number of layers, $L$, the number of spatial wavelet scales, $J_s$, and the number of temporal wavelet scales, $J_t$, are represented by $(J_s, J_t, L)$ for separable ST-GST, and $(J, L)$ for joint ST-GST. Training ratio means the fraction of data used for training in the training set. For the spatial domain, we use the skeleton graph; and for the temporal domain, we use a line graph connecting consecutive time stamps, see Fig. 1(a)(b). Geometric scattering wavelets are used

in both domain, and the nonlinear activation $\sigma(\cdot)$ is absolute value function which has the property of energy-preserving. Features output by ST-GST are then utilized by random forest classifier with 300 decision trees for classification.

**Comparison with state-of-the-art methods.** We consider two datasets, MSR Action3D and NTU-RGB+D (cross-subject). For MSR Action3D, the proposed ST-GST is compared with GFT facilitated by temporal pyramid matching (GFT+TPM) (Kao et al., 2019), Bayesian hierarchical dynamic model (HDM) (Zhao et al., 2019), and a few deep learning approaches, including temporal convolution neural networks (Kim & Reiter, 2017), ST-GCN (Yan et al., 2018), and MS-G3D (Liu et al., 2020). For NTU-RGB+D, Deep LSTM (Liu et al., 2019), part-aware LSTM (PA-LSTM) (Liu et al., 2019) and spatio-temporal LSTM with trust gates (ST-LSTM+TG) (Liu et al., 2016) are included in comparison. Methods labeled "fixed topology" are modified so as not to use adaptive training of the adjacency matrix in order for the comparison with ST-GST to be fair. Tables 1 and 2 compares the classification accuracies on MSR Action3D and NTU-RGB+D, respectively. We see that even without any training, the performance of ST-GST is better than other non-deep-learning and LSTM-based methods, and is comparable with state-of-the-art GCN-based methods in large-scale dataset. Further, ST-GST outperforms all other methods when the size of training set is small. Fig. 3(a) shows the classification accuracy as a function of the training ratio. When training ratio is less than 20%, ST-GST significantly outperforms ST-GCN. Fig. 3(b) shows the accuracy-running time plot, reflecting that ST-GST is much faster than ST-GCN with similar classification performance.

**ST-GST works well in small-scale-data regime.** Table 1 and Fig. 3(a) show that ST-GST outperforms other deep learning methods in the small-scale-data regime, which can be explained as follows. The good performance of spatio-temporal graph neural networks highly relies on the assumption that the training data is abundant. When the size of training set is limited, most of them can be easily trapped into bad local optima due to overfitting, resulting in a significant drop of classification accuracy. But in practice, obtaining a huge amount of training data with high-quality labels could be extremely expensive. On the other hand, since ST-GST is a non-trainable framework, filter coefficients in ST-GST are mathematically designed rather than trained by data, which avoids the problem of overfitting when the training ratio is low. Another advantage of ST-GST compared to ST-GCN is that it requires less computation because no training process is involved in ST-GST.

**Separable design is better than joint design.** Tables 1 and 2 also show that separable spatio-temporal graph wavelets work much better than joint ones, achieving $25\%$ increase in classification accuracy for MSR Action3D dataset. The result is consistent with our analysis in Section 3.2. The intuition is that when dealing with spatio-temporal data generated from complex structures like skeleton sequences, the fixed dependencies generated by product graphs highly restrict the way how signals can be diffused in spatio-temporal graphs and thus limit the efficiency of feature extraction.

**Nonlinearity is beneficial.** Fig. 3(c) compares ST-GST with and without nonlinearity and shows that it is critical to ST-GST, also reflecting the potential effect of nonlinearity in ST-GCNs.

## 6 CONCLUSIONS

In this work we propose a novel spatio-temporal graph scattering transform (ST-GST), which can be viewed as one forward pass of spatio-temporal graph convolutional networks (ST-GCNs) without any training. ST-GST is stable to small perturbations on both input signals and structures. Our experiments show that: i) ST-GST can achieve better performance than both non-deep-learning and ST-GCNs based methods when the size of training samples is limited; ii) designing spatial and temporal graph filters separately is more flexible and computationally efficient than designing them jointly; and iii) the nonlinearity is critical to the performance.

## 7 ACKNOWLEDGEMENT

This work is fully supported by Mitsubishi Electric Research Laboratories (MERL), where Chao Pan was a research intern, Siheng Chen was a research scientist and Antonio Ortega is a consultant.

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

## A  DIFFERENT DESIGN OF GRAPH WAVELETS

There are many off-the-shelf, well-developed graph wavelets we can choose. They mainly focus on extracting features from multiple frequency bands of input signal spectrum. Some of them are shown as follows.

**Monic Cubic wavelets.** Monic Cubic wavelets (Hammond et al., 2011) define the kernel function $h(\lambda)$ as

$$h(\lambda) = \begin{cases} \lambda & \text{for} \quad \lambda < 1; \\ -5 + 11\lambda - 6\lambda^2 + \lambda^3 & \text{for} \quad 1 \leq \lambda \leq 2; \\ 2/\lambda & \text{for} \quad \lambda > 2. \end{cases}$$

Different scales of filters are implemented by scaling and translation of above kernel function.

**Itersine wavelets.** Itersine wavelets define the kernel function at scale $j$ as

$$h_j(\lambda) = \sin\left(\frac{\pi}{2}\cos^2(\pi(\lambda - \frac{j-1}{2}))\right) \mathbb{1}\left[\frac{j}{2} - 1 \leq \lambda \leq \frac{j}{2}\right].$$

Itersine wavelets form tight frames.

**Geometric scattering wavelets.** Geometric scattering wavelet filter bank (Gao et al., 2019) contains a set of filters based on lazy random walk matrix. The filter at scale $j$ is defined as $\mathbf{H}_j(\mathbf{S}) = \mathbf{S}^{2^{j-1}} - \mathbf{S}^{2^j} = \mathbf{S}^{2^{j-1}}(\mathbf{I} - \mathbf{S}^{2^{j-1}})$, where $\mathbf{S} = \frac{1}{2}(\mathbf{I} + \mathbf{A}\mathbf{D}^{-1})$ is the lazy random walk matrix and $\mathbf{D}$ is the degree matrix.

Note that one is also allowed to customize either spatial or temporal graph wavelets, once they conform a frame and satisfy integral Lipschitz constraint shown as follows

$$A^2\|\mathbf{x}\|^2 \leq \sum_{j=1}^{J}\|\mathbf{H}_j\mathbf{x}\|^2 \leq B^2\|\mathbf{x}\|^2, \quad |\lambda h'(\lambda)| \leq \text{const} \; \forall\lambda,$$

where $A, B$ are scalar constants and $h'(\cdot)$ is the gradient of the kernel function.

## B  PROOFS

### B.1  PROOF OF LEMMA 1

By reshaping the signal from $\mathbf{Z}$ to $\mathbf{z}$ with $\mathbf{Z}_{s,t} = \mathbf{z}_{(s-1)T+t}$, we can have that

$$\sum_{j_1,j_2=1}^{J_s,J_t}\|(\mathbf{H}_{j_1}(\mathbf{S}_s) \otimes \mathbf{G}_{j_2}(\mathbf{S}_t))\mathbf{z}\|^2 = \sum_{j_1,j_2=1}^{J_s,J_t}\left\|\mathbf{H}_{j_1}(\mathbf{S}_s)\mathbf{Z}\mathbf{G}_{j_2}^\top(\mathbf{S}_t)\right\|^2.$$

Since $\mathbf{S}_s$ and $\mathbf{S}_t$ do not change over computation process, we just use $\mathbf{H}_{j_1}$ and $\mathbf{G}_{j_2}$ to represent $\mathbf{H}_{j_1}(\mathbf{S}_s)$ and $\mathbf{G}_{j_2}(\mathbf{S}_t)$, respectively. Suppose $\mathbf{H}_{j_1} = \begin{pmatrix} h_{11} & & h_{1N} \\ & \ddots & \\ h_{N1} & & h_{NN} \end{pmatrix} \in \mathbb{R}^{N \times N}$, then we have

the Kronecker product as $\mathbf{H}_{j_1} \otimes \mathbf{G}_{j_2} = \begin{pmatrix} h_{11}\mathbf{G}_{j_2} & & h_{1N}\mathbf{G}_{j_2} \\ & \ddots & \\ h_{N1}\mathbf{G}_{j_2} & & h_{NN}\mathbf{G}_{j_2} \end{pmatrix}$. Apply it to vector $\mathbf{z}$ and we can have a filtered signal $\mathbf{y}_{j_1,j_2} = (\mathbf{H}_{j_1} \otimes \mathbf{G}_{j_2})\mathbf{z} \in \mathbb{R}^{NT}$. The first $T$ elements of $\mathbf{y}$ can also be written as

$$\mathbf{y}_{j_1,j_2}(1:T) = \sum_{i=1}^{N} h_{1i}\mathbf{G}_{j_2}\begin{pmatrix}\mathbf{Z}_{i,1}\\\mathbf{Z}_{i,2}\\\vdots\\\mathbf{Z}_{i,T}\end{pmatrix} = \mathbf{G}_{j_2}\sum_{i=1}^{N} h_{1i}\begin{pmatrix}\mathbf{Z}_{i,1}\\\mathbf{Z}_{i,2}\\\vdots\\\mathbf{Z}_{i,T}\end{pmatrix}.$$

Therefore we have

$$A_2^2 \left\| \sum_{i=1}^N h_{1i} \begin{pmatrix} \mathbf{Z}_{i,1} \\ \mathbf{Z}_{i,2} \\ \vdots \\ \mathbf{Z}_{i,T} \end{pmatrix} \right\|^2 \leq \sum_{j_2} \|\mathbf{y}_{j_1,j_2}(1:T)\|^2 \leq B_2^2 \left\| \sum_{i=1}^N h_{1i} \begin{pmatrix} \mathbf{Z}_{i,1} \\ \mathbf{Z}_{i,2} \\ \vdots \\ \mathbf{Z}_{i,T} \end{pmatrix} \right\|^2 .$$

Thus $\sum_{j_2} \|\mathbf{y}_{j_1,j_2}\|^2$ can be sandwiched as

$$A_2^2 \sum_{k=1}^N \left\| \sum_{i=1}^N h_{ki} \begin{pmatrix} \mathbf{Z}_{i,1} \\ \mathbf{Z}_{i,2} \\ \vdots \\ \mathbf{Z}_{i,T} \end{pmatrix} \right\|^2 \leq \sum_{j_2} \|\mathbf{y}_{j_1,j_2}\|^2 \leq B_2^2 \sum_{k=1}^N \left\| \sum_{i=1}^N h_{ki} \begin{pmatrix} \mathbf{Z}_{i,1} \\ \mathbf{Z}_{i,2} \\ \vdots \\ \mathbf{Z}_{i,T} \end{pmatrix} \right\|^2 .$$

By definition of vector $\ell_2$ norm, we can rewrite the upper and lower bound in Eq. (6) as

$$A_2^2 \sum_{i=1}^T \left\| \mathbf{H}_{j_1} \begin{pmatrix} \mathbf{Z}_{1,i} \\ \mathbf{Z}_{2,i} \\ \vdots \\ \mathbf{Z}_{N,i} \end{pmatrix} \right\|^2 \leq \sum_{j_2} \|\mathbf{y}_{j_1,j_2}\|^2 \leq B_2^2 \sum_{i=1}^T \left\| \mathbf{H}_{j_1} \begin{pmatrix} \mathbf{Z}_{1,i} \\ \mathbf{Z}_{2,i} \\ \vdots \\ \mathbf{Z}_{N,i} \end{pmatrix} \right\|^2 .$$

Summing above quantity over $j_1$ gives us that

$$A_1^2 A_2^2 \|\mathbf{Z}\|^2 = A_1^2 A_2^2 \sum_{i=1}^T \left\| \begin{pmatrix} \mathbf{Z}_{1,i} \\ \mathbf{Z}_{2,i} \\ \vdots \\ \mathbf{Z}_{N,i} \end{pmatrix} \right\|^2 \leq \sum_{j_1,j_2} \|\mathbf{y}_{j_1,j_2}\|^2 \leq B_1^2 B_2^2 \sum_{i=1}^T \left\| \begin{pmatrix} \mathbf{Z}_{1,i} \\ \mathbf{Z}_{2,i} \\ \vdots \\ \mathbf{Z}_{N,i} \end{pmatrix} \right\|^2 = B_1^2 B_2^2 \|\mathbf{Z}\|^2 ,$$

which completes the proof. Lemma 1 is a very handful result. It shows that we can easily construct new spatio-temporal wavelets just by combining spatio and temporal ones. Moreover, the constants for new frame bound can be easily obtained once we know the characteristics of the wavelets in each domain. In particular, it also provides us a convenient way to build tight frames for spatio-temporal data analysis with $A = B$, because we just need to choose tight frames for spatial and temporal domain separately without considering possible correlations.

## B.2 PROOF OF THEOREM 1

We are considering pooling operator $U(\cdot)$ as average in spatial domain in this proof, so $\mathbf{U} = \frac{1}{N} \mathbf{1}_{1 \times N}$ and $\phi = \mathbf{U}\mathbf{Z} \in \mathbb{R}^T$. The proof techniques can be easily generalized to any form of $U(\cdot)$. When reshaping $\mathbf{Z} \in \mathbb{R}^{N \times T}$ to $\mathbf{z} \in \mathbb{R}^{NT}$, the new pooling operator can be simply represented as

$$\mathbf{U}' = \frac{1}{N}(\mathbf{I}_T, \mathbf{I}_T, \cdots, \mathbf{I}_T) \in \mathbb{R}^{T \times NT}, \quad \phi = \mathbf{U}'\mathbf{z}.$$

Note that $\|\mathbf{U}'\|_2 = \frac{1}{\sqrt{N}}$. Consider scattering tree nodes at the last layer $L-1$. Suppose they are indexed from 1 to $J^{L-1}$ associated with signal $\mathbf{a}_1, \cdots, \mathbf{a}_{J^{L-1}}$, and their parent nodes are indexed from 1 to $J^{L-2}$ associated with signal $\mathbf{b}_1, \cdots, \mathbf{b}_{J^{L-2}}$. When the input data $\mathbf{X}$ is perturbed, all signals in scattering tree will change correspondingly. Here we simply denote them as $\widetilde{\mathbf{a}}, \widetilde{\mathbf{b}}$. Then for the change of feature vector located at node with $\mathbf{a}_1$, it holds that

$$\|\phi_{\mathbf{a}_1} - \phi_{\widetilde{\mathbf{a}}_1}\|^2 = \|\mathbf{U}'(\mathbf{a}_1 - \widetilde{\mathbf{a}}_1)\|^2 \leq \|\mathbf{U}'\|^2 \|\mathbf{a}_1 - \widetilde{\mathbf{a}}_1\|^2 \leq \frac{1}{N} \|\sigma((\mathbf{H}_{j_1} \otimes \mathbf{G}_{j_2})(\mathbf{b}_1 - \widetilde{\mathbf{b}}_1))\|^2, \quad (6)$$

where $j_1 = j_2 = 1$. The last inequality holds because we are using absolute value function as nonlinear activation, which is non-expansive. Summing above quantity over $j_1, j_2$ and by the frame bound proved in Lemma 1, we can have that

$$\sum_{i=1}^{J^{L-1}} \|\phi_{\mathbf{a}_i} - \phi_{\widetilde{\mathbf{a}}_i}\|^2 \leq \frac{B^2}{N} \sum_{i=1}^{J^{L-2}} \|\mathbf{b}_i - \widetilde{\mathbf{b}}_i\|^2. \quad (7)$$

Note that for sum of square norm of change at layer $L-2$ it is

$$\sum_{i=1}^{J^{L-2}} \|\boldsymbol{\phi}_{\mathbf{b}_i} - \boldsymbol{\phi}_{\widetilde{\mathbf{b}}_i}\|^2 \leq \frac{1}{N} \sum_{i=1}^{J^{L-2}} \|\mathbf{b}_i - \widetilde{\mathbf{b}}_i\|^2. \tag{8}$$

Compare Eq. (7) and (8). The upper bound only differs with a factor $B^2$. Then by induction we can have that

$$\|\Phi(\mathbf{S}_s, \mathbf{S}_t, \mathbf{X}) - \Phi(\mathbf{S}_s, \mathbf{S}_t, \widetilde{\mathbf{X}})\|^2 \leq \frac{1}{N} \sum_{\ell=0}^{L-1} B^{2\ell} \|\mathbf{x} - \widetilde{\mathbf{x}}\|^2 = \frac{1}{N} \sum_{\ell=0}^{L-1} B^{2\ell} \|\boldsymbol{\Delta}\|^2.$$

Normalize it with the dimension of final feature map, we have

$$\frac{\|\Phi(\mathbf{S}_s, \mathbf{S}_t, \mathbf{X}) - \Phi(\mathbf{S}_s, \mathbf{S}_t, \widetilde{\mathbf{X}})\|}{\sqrt{T \sum_{\ell=0}^{L-1} J^\ell}} \leq \frac{1}{\sqrt{NT}} \sqrt{\frac{\sum_{\ell=0}^{L-1} B^{2\ell}}{\sum_{\ell=0}^{L-1} J^\ell}} \|\boldsymbol{\Delta}\|. \tag{9}$$

### B.3 Proof of Theorem 2

Perturbations on the underlying graph usually happen when the graph is unknown or when the graph changes over time (Segarra et al., 2017). Take skeleton-based action recognition as an example. Some joints may be misrecognized with others due to measurement noise of devices during certain frames, thus the location signals of those joints are interchanged. This leads to different spatial graph structures at those time stamps. Since such kind of perturbations usually happen in spatial domain, here we simply consider the structure perturbations on the spatial graph only. But the results can be extended to more general cases.

Consider the original spatio-temporal graph as $\mathcal{G}$ with spatial graph shift matrix $\mathbf{S}_s$ and temporal one $\mathbf{S}_t$, and the perturbed graph as $\widehat{\mathcal{G}}$ with $\widehat{\mathbf{S}}_s$ and $\mathbf{S}_t$. We first show that ST-GST is invariant to node permutations in spatial domain, where the set of permutation matrices is defined as $\mathcal{P} = \{\mathbf{P} \in \{0,1\}^{N \times N} : \mathbf{P}\mathbf{1} = 1, \mathbf{P}^\top\mathbf{1} = 1, \mathbf{P}\mathbf{P}^\top = \mathbf{I}_N\}$. Note that we are considering average in spatial domain for $U(\cdot)$, so $\mathbf{U} = \frac{1}{N}\mathbf{1}_{1 \times N}$ and $\boldsymbol{\phi} = \mathbf{U}\mathbf{Z} \in \mathbb{R}^T$, $\widehat{\mathbf{U}} = \mathbf{U}\mathbf{P}$.

**Lemma 2.** Consider the spatial permutation $\widehat{\mathbf{S}}_s = \mathbf{P}^\top\mathbf{S}_s\mathbf{P}$ and input data $\widehat{\mathbf{X}} = \mathbf{P}^\top\mathbf{X}$ are also permuted in spatial domain correspondingly. Then, it holds that

$$\Phi(\mathbf{S}_s, \mathbf{S}_t, \mathbf{X}) = \Phi(\widehat{\mathbf{S}}_s, \mathbf{S}_t, \widehat{\mathbf{X}}) \tag{10}$$

*Proof.* Note that the permutation holds for all signals computed in scattering tree; that is to say, $\widehat{\mathbf{Z}}_{(p^{(\ell)})} = \mathbf{P}^\top\mathbf{Z}_{(p^{(\ell)})}$. Suppose for path $p^{(\ell)}$ the last two filter are chosen as $\mathbf{H}(\widehat{\mathbf{S}}_s)$ and $\mathbf{G}(\mathbf{S}_t)$, then the feature vector after pooling with respect to new graph support and data can be written as

$$\begin{aligned}
\boldsymbol{\phi}_{(p^{(\ell)})}(\widehat{\mathbf{S}}_s, \mathbf{S}_t, \widehat{\mathbf{Z}}_{(p^{(\ell)})}) &= \widehat{\mathbf{U}}(\sigma(\mathbf{H}(\widehat{\mathbf{S}}_s)\widehat{\mathbf{Z}}_{(p^{(\ell)})}\mathbf{G}^\top(\mathbf{S}_t))) \\
&= \mathbf{U}\mathbf{P}\sigma(\mathbf{P}^\top\mathbf{H}(\mathbf{S}_s)\mathbf{P}\mathbf{P}^\top\mathbf{Z}_{(p^{(\ell)})}\mathbf{G}^\top(\mathbf{S}_t))
\end{aligned}$$

The last equation holds due to definition of $\mathbf{H}(\mathbf{S})$. Since nonlinear activation is applied element-wise, we can rewrite it as

$$\begin{aligned}
\boldsymbol{\phi}_{(p^{(\ell)})}(\widehat{\mathbf{S}}_s, \mathbf{S}_t, \widehat{\mathbf{Z}}_{(p^{(\ell)})}) &= \mathbf{U}\sigma(\mathbf{P}\mathbf{P}^\top\mathbf{H}(\mathbf{S}_s)\mathbf{P}\mathbf{P}^\top\mathbf{Z}_{(p^{(\ell)})}\mathbf{G}^\top(\mathbf{S}_t)) \\
&= \mathbf{U}\sigma(\mathbf{H}(\mathbf{S}_s)\mathbf{Z}_{(p^{(\ell)})}\mathbf{G}^\top(\mathbf{S}_t)) \\
&= \boldsymbol{\phi}_{(p^{(\ell)})}(\mathbf{S}_s, \mathbf{S}_t, \mathbf{Z}_{(p^{(\ell)})}).
\end{aligned}$$

This conclusion holds independently of specific path $p^{(\ell)}$, so it holds for all feature vector after pooling in scattering tree. Since final feature map is just a concatenation of all feature vectors, the proof is complete. □

Lemma 2 shows that the output of ST-GST is essentially independent of the node ordering in spatial domain, as long as the permutation is consistent across all time stamps. This result is intuitive

because the output of graph convolution should only depend on relative neighborhood structure of each node. Since node reordering will not alter neighborhood topology, the output should remain unchanged.

Based on Lemma 2, we use a relative perturbation model for structure modifications (Gama et al., 2019b), which focuses more on the change of neighborhood topology compared to absolute perturbations adopted in Levie et al. (2019). Define the set of permutations that make $\mathbf{S}_s$ and $\widehat{\mathbf{S}}$ the closet as $\mathcal{P}_s := \arg\min_{\mathbf{P} \in \mathcal{P}} \|\mathbf{P}^\top \widehat{\mathbf{S}}_s \mathbf{P} - \mathbf{S}_s\|_2$. Consider the set of perturbation matrices $\mathcal{E}(\mathbf{S}, \widehat{\mathbf{S}}) = \{\mathbf{E} | \mathbf{P}^\top \widehat{\mathbf{S}}_s \mathbf{P} = \mathbf{S}_s + \mathbf{E}^\top \mathbf{S}_s + \mathbf{S}_s \mathbf{E}, \mathbf{P} \in \mathcal{P}_s, \mathbf{E} \in \mathbb{R}^{N \times N}\}$. Then the relative distance to measure structure perturbations can be defined as

$$d(\mathbf{S}_s, \widehat{\mathbf{S}}_s) = \min_{\mathbf{E} \in \mathcal{E}(\mathbf{S}_s, \widehat{\mathbf{S}}_s)} \|\mathbf{E}\|_2$$

Note that if $\widehat{\mathbf{S}}_s = \mathbf{P}^\top \mathbf{S}_s \mathbf{P}$, meaning that the structure perturbation is purely permutation, then the relative distance $d(\mathbf{S}_s, \widehat{\mathbf{S}}_s) = 0$, which is consistent with result shown in Lemma 2. Therefore, without loss of generality, we can assume that $\mathbf{P} = \mathbf{I}_N$ and $\widehat{\mathbf{S}}_s = \mathbf{S}_s + \mathbf{E}^\top \mathbf{S}_s + \mathbf{S}_s \mathbf{E}$ in later context. With this formulation, we are ready to prove Lemma 3.

**Lemma 3.** Suppose eigenvalues $\{m_i\}_{i=1}^N$ of $\mathbf{E}$ are organized in order such that $|m_1| \leq |m_2| \leq \cdots \leq |m_N|$, satisfying $|m_N| \leq \epsilon/2$ and $|m_i/m_N - 1| \leq \epsilon$ for $\epsilon > 0$. For spatial graph filter $\mathbf{H}(\mathbf{S}_s)$ and temporal graph filter $\mathbf{G}(\mathbf{S}_t)$, denote their kernel functions as $h(\lambda)$ and $g(\lambda)$, respectively. If for all $\lambda$, $h(\lambda)$ is chosen to satisfy integral Lipschitz constraint $|\lambda h'(\lambda)| \leq C$ and $g(\lambda)$ has bounded spectral response $|g(\lambda)| \leq D$. Then it holds that

$$\|\mathbf{H}(\mathbf{S}_s) \otimes \mathbf{G}(\mathbf{S}_t) - \mathbf{H}(\widehat{\mathbf{S}}_s) \otimes \mathbf{G}(\mathbf{S}_t)\|_2 \leq \epsilon CD + O(\epsilon^2). \tag{11}$$

*Proof.* From Proposition 2 in Gama et al. (2019b) we can have that when $\mathbf{E}$ satisfies above conditions, $\|\mathbf{H}(\mathbf{S}_s) - \mathbf{H}(\widehat{\mathbf{S}}_s)\|_2 \leq \epsilon C + O(\epsilon^2)$. So

$$\begin{aligned}
\|\mathbf{H}(\mathbf{S}_s) \otimes \mathbf{G}(\mathbf{S}_t) - \mathbf{H}(\widehat{\mathbf{S}}_s) \otimes \mathbf{G}(\mathbf{S}_t)\|_2 &= \|(\mathbf{H}(\mathbf{S}_s) - \mathbf{H}(\widehat{\mathbf{S}}_s)) \otimes \mathbf{G}(\mathbf{S}_t)\|_2 \\
&\leq \|\mathbf{H}(\mathbf{S}_s) - \mathbf{H}(\widehat{\mathbf{S}}_s)\|_2 \|\mathbf{G}(\mathbf{S}_t)\|_2 \\
&\leq \epsilon CD + O(\epsilon^2),
\end{aligned}$$

The second line holds because $\mathbf{H}(\mathbf{S}_s) - \mathbf{H}(\widehat{\mathbf{S}}_s)$ is a symmetric matrix, which can be written as eigen-decomposition as $\mathbf{F}\boldsymbol{\Omega}\mathbf{F}^\top$. And $(\mathbf{F}\boldsymbol{\Omega}\mathbf{F}^\top) \otimes (\mathbf{V}\boldsymbol{\Lambda}\mathbf{V}^T) = (\mathbf{F} \otimes \mathbf{V})(\boldsymbol{\Omega} \otimes \boldsymbol{\Lambda})(\mathbf{F} \otimes \mathbf{V})^\top$ holds, which finishes the proof. As for general structural perturbations, where we want to find $\|\mathbf{H}(\mathbf{S}_s) \otimes \mathbf{G}(\mathbf{S}_t) - \mathbf{H}(\widehat{\mathbf{S}}_s) \otimes \mathbf{G}(\widehat{\mathbf{S}}_t)\|_2$, we can add and subtract term $\mathbf{H}(\widehat{\mathbf{S}}_s) \otimes \mathbf{G}(\widehat{\mathbf{S}}_t)$, use triangle inequality and further bound those two terms with more assumptions on $h(\lambda)$ and $g(\lambda)$. $\square$

The bound shown in Lemma 3 indicates that the difference of output caused by changing spatial graph support from $\mathbf{S}_s$ to $\widehat{\mathbf{S}}_s$ is proportional to $\epsilon$, which is a scalar characterizing the level of the perturbation. Constraints on eigenvalues of $\mathbf{E}$ limits the change of graph structure. A more detailed description explaining the necessity of such constraints can be found in Gama et al. (2019b). With Lemma 3 in hand, we are ready to show the change of feature vector after pooling at each node in scattering tree when such structure perturbations happen.

**Lemma 4.** Consider a ST-GST with $L$ layers and $J = J_s \times J_t$ scales at each layer. Suppose that the graph filter bank forms a frame with upper bound $B = B_1 \times B_2$, where $B_1, B_2$ are frame bounds for spatial and temporal domain, respectively. Suppose for all $\lambda$, spatial wavelet filter bank $\{\mathbf{H}_{j_1}\}_{j_1=1}^{J_s}$ satisfies $\max_i |\lambda h_i'(\lambda)| \leq C$ and temporal wavelet filter bank $\{\mathbf{G}_{j_2}\}_{j_2=1}^{J_t}$ satisfies $\max_i |g_i(\lambda)| \leq D$, and other conditions the same as Lemma 3. Then for the change of feature vector $\boldsymbol{\phi}_{p^{(\ell)}}$ associated with path $p^{(\ell)}$ it holds that

$$\|\boldsymbol{\phi}_{p^{(\ell)}}(\mathbf{S}_s, \mathbf{S}_t, \mathbf{X}) - \boldsymbol{\phi}_{p^{(\ell)}}(\widehat{\mathbf{S}}_s, \mathbf{S}_t, \mathbf{X})\| \leq \frac{1}{\sqrt{N}} \epsilon \ell CDB^{\ell-1} \|\mathbf{X}\|. \tag{12}$$

*Proof.* Expand $\|\phi_{p^{(\ell)}}(\mathbf{S}_s, \mathbf{S}_t, \mathbf{X}) - \phi_{p^{(\ell)}}(\widehat{\mathbf{S}}_s, \mathbf{S}_t, \mathbf{X})\|$ as

$$\|\mathbf{U}'(\sigma(\mathbf{H}_{j_1^{(\ell)}}(\mathbf{S}_s) \otimes \mathbf{G}_{j_2^{(\ell)}}(\mathbf{S}_t)))_{p^{(\ell)}}\mathbf{x} - \mathbf{U}'(\sigma(\mathbf{H}_{j_1^{(\ell)}}(\widehat{\mathbf{S}}_s) \otimes \mathbf{G}_{j_2^{(\ell)}}(\mathbf{S}_t)))_{p^{(\ell)}}\mathbf{x}\|$$

$$\leq \frac{1}{\sqrt{N}}\|(\sigma(\mathbf{H}_{j_1^{(\ell)}}(\mathbf{S}_s) \otimes \mathbf{G}_{j_2^{(\ell)}}(\mathbf{S}_t)))_{p^{(\ell)}}\mathbf{x} - (\sigma(\mathbf{H}_{j_1^{(\ell)}}(\widehat{\mathbf{S}}_s) \otimes \mathbf{G}_{j_2^{(\ell)}}(\mathbf{S}_t)))_{p^{(\ell)}}\mathbf{x}\|,$$

where $\|\mathbf{U}'\|_2 = 1/\sqrt{N}$ and $(\sigma(\mathbf{H}_{j_1^{(\ell)}}(\mathbf{S}_s) \otimes \mathbf{G}_{j_2^{(\ell)}}(\mathbf{S}_t)))_{p^{(\ell)}}$ is a shorthand for applying spatio-temporal filters and nonlinear activation in order to input data $\ell$ times according to the path $p^{(\ell)}$. Add and subtract term $\sigma(\mathbf{H}_{j_1^{(\ell)}}(\mathbf{S}_s) \otimes \mathbf{G}_{j_2^{(\ell)}}(\mathbf{S}_t))\sigma(\mathbf{H}_{j_1^{(\ell-1)}}(\widehat{\mathbf{S}}_s) \otimes \mathbf{G}_{j_2^{(\ell-1)}}(\mathbf{S}_t)) \cdots \sigma(\mathbf{H}_{j_1^{(1)}}(\widehat{\mathbf{S}}_s) \otimes \mathbf{G}_{j_2^{(1)}}(\mathbf{S}_t))\mathbf{x}$ and apply triangle inequality, we can have that

$$\|(\sigma(\mathbf{H}_{j_1^{(\ell)}}(\mathbf{S}_s) \otimes \mathbf{G}_{j_2^{(\ell)}}(\mathbf{S}_t)))_{p^{(\ell)}}\mathbf{x} - (\sigma(\mathbf{H}_{j_1^{(\ell)}}(\widehat{\mathbf{S}}_s) \otimes \mathbf{G}_{j_2^{(\ell)}}(\mathbf{S}_t)))_{p^{(\ell)}}\mathbf{x}\|$$

$$\leq \|\sigma(\mathbf{H}_{j_1^{(\ell)}}(\mathbf{S}_s) \otimes \mathbf{G}_{j_2^{(\ell)}}(\mathbf{S}_t))\left((\sigma(\mathbf{H}_{j_1^{(\ell-1)}}(\mathbf{S}_s) \otimes \mathbf{G}_{j_2^{(\ell-1)}}(\mathbf{S}_t)))_{p^{(\ell-1)}} - \right.$$

$$\left. (\sigma(\mathbf{H}_{j_1^{(\ell-1)}}(\widehat{\mathbf{S}}_s) \otimes \mathbf{G}_{j_2^{(\ell-1)}}(\mathbf{S}_t)))_{p^{(\ell-1)}}\right)\mathbf{x}\| +$$

$$\|\left(\sigma(\mathbf{H}_{j_1^{(\ell)}}(\mathbf{S}_s) \otimes \mathbf{G}_{j_2^{(\ell)}}(\mathbf{S}_t)) - \sigma(\mathbf{H}_{j_1^{(\ell)}}(\widehat{\mathbf{S}}_s) \otimes \mathbf{G}_{j_2^{(\ell)}}(\mathbf{S}_t))\right) \cdot$$

$$(\sigma(\mathbf{H}_{j_1^{(\ell-1)}}(\widehat{\mathbf{S}}_s) \otimes \mathbf{G}_{j_2^{(\ell-1)}}(\mathbf{S}_t)))_{p^{(\ell-1)}}\mathbf{x}\|.$$

Recursive quantities can be observed above and the bound can be solved explicitly (Gama et al., 2019b). By induction and conclusion from Lemma 3, we can get that

$$\|(\sigma(\mathbf{H}_{j_1^{(\ell)}}(\mathbf{S}_s) \otimes \mathbf{G}_{j_2^{(\ell)}}(\mathbf{S}_t)))_{p^{(\ell)}}\mathbf{x} - (\sigma(\mathbf{H}_{j_1^{(\ell)}}(\widehat{\mathbf{S}}_s) \otimes \mathbf{G}_{j_2^{(\ell)}}(\mathbf{S}_t)))_{p^{(\ell)}}\mathbf{x}\| \leq \ell\epsilon CDB^{\ell-1}\|\mathbf{x}\|.$$

Multiplying the coefficient $1/\sqrt{N}$ caused by pooling gets us the final result. □

Note that the upper bound in Lemma 4 holds for all path of length $\ell$. Thus the square norm of change in final feature map can be summarized by the sum of square norm of change at each layer, which finishes the proof of Theorem 2.

## C  ADDITIONAL EXPERIMENTS

### C.1  DATASET

**MSR Action3D dataset** (Li et al., 2010) is a small dataset capturing indoor human actions. It covers 20 action types and 10 subjects, with each subject repeating each action 2 or 3 times. The dataset contains 567 action clips with maximum number of frames 76; however, 10 of them are discarded because the skeleton information are either missing or too noisy (Wang et al., 2012). For each clip, locations of 20 joints are recorded, and only one subject is present. Training and testing set is decided by cross-subject split for this dataset, with 288 samples for training and 269 for testing.

**NTU-RGB+D** (Liu et al., 2019) is currently the largest dataset with 3D joints annotations for human action recognition task. It covers 60 action types and 40 subjects. The dataset contains 56,880 action clips with maximum number of frames 300, and there are 25 joints for each subject in one clip. Each clip is guaranteed to have at most 2 subjects. The cross-subject benchmark of NTU-RGB+D includes 40,320 clips for training and 16,560 for testing.

**Full table of performance on MSR Action3D dataset**. The table contains performance comparison for different algorithms with different set of parameters on MSR Action3D dataset. Note that the triple shown after ST-GST represents the value for $(J_s, J_t, L)$. Methods labeled "fixed topology" are modified so as not to use adaptive training of the adjacency matrix in order for the comparison with ST-GST to be fair. Methods labeled "learnable topology" means that we use adaptive training for adjacency matrix to further validate our claim. Other configurations of compared methods are then set by default. From the table we can see that ST-GST outperforms all other methods even when the graph topology can be learned by neural networks. The intuition behind this is that deep learning methods need large amount of training data due to the complex structures, and it can easily

| | Method | Accuracy (%) |
|---|---|---|
| | GFT+TPM | 74.0 |
| | HDM | 81.8 |
| GNNs | ST-GCN (fixed topology) | 52.0 |
| | ST-GCN (learnable topology) | 56.0 |
| | Temporal Conv. (resnet) | 67.3 |
| | Temporal Conv. (resnet-v3-gap) | 69.9 |
| | Temporal Conv. (resnet-v4-gap) | 72.1 |
| | MS-G3D (GCN scales=10, G3D scales=6) | 80.3 |
| | MS-G3D (GCN scales=5, G3D scales=5) | 81.4 |
| | MS-G3D (GCN scales=8, G3D scales=5) | 82.2 |
| Scattering | Separable ST-GST (5, 5, 3) | 73.6 |
| | Separable ST-GST (5, 5, 4) | 72.9 |
| | Separable ST-GST (5, 10, 3) | 81.4 |
| | Separable ST-GST (5, 15, 3) | 85.9 |
| | Separable ST-GST (5, 20, 3) | **87.0** |
| | Joint Kronecker ST-GST (15, 3) | 61.0 |
| | Joint Cartesian ST-GST (15, 3) | 59.1 |
| | Joint Strong ST-GST (15, 3) | 61.7 |

Table 3: Full comparison of classification accuracy (MSR Action3D with 288 training and 269 testing samples).

| Method | Accuracy (%) |
|---|---|
| Separable ST-GST (5, 5, 3) | $73.4 \pm 0.8$ |
| Separable ST-GST (5, 20, 3) | $86.7 \pm 0.4$ |
| Joint Kronecker ST-GST (5, 3) | $46.3 \pm 1.2$ |
| Joint Cartesian ST-GST (5, 3) | $42.2 \pm 1.1$ |
| Joint Strong ST-GST (5, 3) | $45.0 \pm 1.2$ |
| Joint Kronecker ST-GST (15, 3) | $59.6 \pm 0.5$ |
| Joint Cartesian ST-GST (15, 3) | $58.6 \pm 1.0$ |
| Joint Strong ST-GST (15, 3) | $60.0 \pm 1.0$ |

Table 4: Performance for different methods on MSR Action3D with standard deviations.

be trapped into bad local optima due to overfitting when the size of training set is limited, which is common in practice. Also the good performance of ST-GST in sparse label regime could potentially inspire active learning for processing spatio-temporal data (Bilgic et al., 2010).

**Performance on MSR Action3D dataset with standard deviations.** We repeat part of our experiments 20 times on MSR Action3D dataset, especially for joint approaches, to obtain the standard deviations of classification accuracy. The results are shown in Table 4. Note that since ST-GST is a mathematically designed transform, the output features should be the same for different trails, and the randomness comes from classifiers used later (random forest in this case). It can be seen that the standard deviations are comparable in all these methods, and therefore the conclusion that separable ST-GST consistently outperforms joint ST-GST still holds.

**Comparison between different choices of wavelets.** In practice we find that using graph geometric scattering wavelets (Gao et al., 2019) for both spatial and temporal domain can achieve the best performance, which is reported in main text. Classification accuracy using other type of wavelets is shown here. All experiments performed here are separable ST-GST with $J_s = 5, J_t = 15, L = 3$ on MSR Action3D dataset. An interesting observation is that there is a significant reduction in accuracy when we change temporal wavelet from diffusion based one (Geometric) to spectrum based one (MonicCubic or Itersine). This may caused by the design of different wavelets.

**Stability of ST-GST.** We also show the classification accuracy under different level of perturbations on spatio-temporal signals and spatial graph structures in Fig. 4. The experiments are con-

| Spatial wavelet | Temporal wavelet | Accuracy (%) |
|---|---|---|
| Geometric | Geometric | 85.9 |
| Geometric | MonicCubic | 76.6 |
| Geometric | Itersine | 73.6 |
| MonicCubic | Geometric | 82.9 |
| Itersine | Geometric | 82.5 |
| MonicCubic | MonicCubic | 80.7 |
| MonicCubic | Itersine | 78.4 |
| Itersine | MonicCubic | 76.2 |
| Itersine | Itersine | 80.7 |

Table 5: Performance for different choices of spatial and temporal wavelets (MSR Action3D) with setting (5, 15, 3).

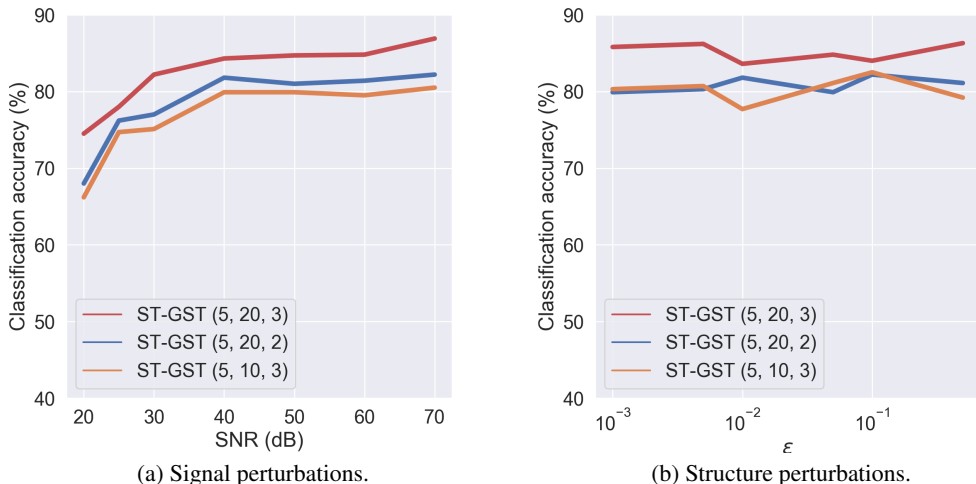

(a) Signal perturbations.       (b) Structure perturbations.

Figure 4: Comparisons on performance under different level of perturbations.

ducted on MSR Action3D dataset. For signal perturbation, signal-to-noise ratio (SNR) is defined as $10 \log \frac{\|\mathbf{X}\|^2}{\|\mathbf{\Delta}\|^2}$. For structure perturbation, $\mathbf{E}$ is set to be a diagonal matrix, whose diagonal elements satisfy corresponding constraints on $\epsilon$. From both Fig. 4(a) and (b) we can see that ST-GST is stable and will not deviate much from original output when the perturbations are small.

