# OpenReview forum: "Spatio-Temporal Graph Scattering Transform"
_ICLR.cc/2021/Conference — ICLR 2021 Poster_

### Official Review · AnonReviewer3 · 2020-10-21
**The Joint Kronecker, Joint Cartesian and Joint Strong, have not achieved the satisfied performance**

**Rating:** 6
**Confidence:** 5

**Review:**

The authors propose wavelets for both separable and joint spatio-temporal graphs. And then the authors design a spatio-temporal graph scattering transform (ST-GST), which is a non-trainable counterpart of spatio-temporal graph convolutional networks and a nonlinear version of spatiotemporal graph wavelets. Finally, the proposed SF-GST is conducted by experiments, and the results show that it appears to be effective. However, The authors did not give the explanation of the motivation about why did the STG should be scattered by wavelets. Besides, from the results in Table 1,  the joint versions based on the proposed method. i.e., Joint Kronecker, Joint Cartesian and Joint Strong, have not achieved the satisfied performance, though only separable versions performs best.

---

> ### Author Response · Authors · 2020-11-16
> **Response to the questions raised by Reviewer 3**
>
> We thank Reviewer 3 for her/his valuable comments and insightful questions for improving the manuscript. We addressed all the comments as described below, and will include this discussion into our revision.
>
> (Q1) "Motivation of using wavelets to analyze spatio-temporal graph signals"
>
> Here is our motivation for adopting wavelets into the framework. To design our spatio-temporal graph scattering transform, we need spatio-temporal graph filter banks at each scattering layer, which do not exist before. Meanwhile, for time-series, wavelets are one of the best tools to design filter banks, allowing us to trade-off between the time-frequency resolutions and touching the lower bound of uncertainty principle of the time-frequency representations. Inspired by this, we choose to expand the wavelet techniques to the spatio-temporal graph domain and efficiently design spatio-temporal graph filter banks based on wavelets.
>
> (Q2) "The performance of joint ST-GST is not comparable to separable ST-GST"
>
> As for the experimental results, we actually expect that the performance of joint ST-GST is not comparable to that of separable ST-GST based on our mathematical derivations, which is also the point we are highlighting in the paper. As stated at the end of Section 3.2, separable ST-GST is more flexible in design. The flexibility of separable ST-GST means that we can design different graph filters ($\mathbf{h}$ and $\mathbf{g}$ in Eq.(1)) with independent reception fields ($P$ and $Q$ in Eq.(1)) in the spatial and temporal domains. On the other hand, for joint graph filters, filter coefficients and the reception fields in the spatial and temporal domains are bonded together ($\mathbf{h}$ and $K$ in Eq.(2));  that is, we cannot change the filter or reception field in the spatial domain without changing the corresponding one in the temporal domain. This thus makes the separable graph filters more flexible than the joint graph filters. Moreover, we can choose different number of scales/filters for spatial and temporal domains in separable ST-GST, which is not achievable in joint ST-GST.
>
> In short, separable ST-GST is more recommended to use in practice because of its flexibility in design, computation efficiency and straightforward interpretation.

---

### Official Review · AnonReviewer2 · 2020-10-28
**Good extension of scattering transform to the spatio-temporal domain**

**Rating:** 7
**Confidence:** 3

**Review:**

The authors extend scattering transform to the spatio-temporal domain. The result ST-GST performs well with small size dataset.

Pros:
1.  The authors propose a novel spatio-temporal graph scattering transform.
2. The authors empirically shows that ST-GST outperform spatio-temporal graph convolutional networks and other non-deep methods in small-scale datasets.


Questions:
1. We know that scattering-transform cannot do as well as CNN on larger dataset like Imagenet. Does ST-GST also have similar limitations?

---

> ### Author Response · Authors · 2020-11-16
> **Response to the questions raised by Reviewer 2**
>
> We thank Reviewer 2 for her/his valuable comments for improving the manuscript. We addressed all the comments as described below.
>
> As shown in Table 2, in large-scale dataset where the training data is **abundant**, ST-GST can outperform many simple deep learning methods, such as deep LSTM, PA-LSTM and ST-LSTM+TG; however, it can hardly outperform the SOTA networks, such as ST-GCN. But in practice, obtaining a huge amount of training data with high-quality labels could be extremely expensive. Most spatio-temporal graph neural network based methods can be easily trapped into bad local optima due to overfitting when the size of training set is limited, resulting in a significant drop of classification accuracy. On the other hand, since ST-GST is a non-trainable framework, filter coefficients are mathematically designed instead of trained by data, so ST-GST can perform much better than ST-GCN when the training ratio (fraction of data used for training in training set) is low. For example, Figure 3(a) shows that ST-GST outperforms ST-GCN by $15\\%$ when training ratio is $5\\%$ for NTU-RGB+D dataset. Another advantage of ST-GST compared to ST-GCN is that it requires less computation resources because no training process is involved in ST-GST. Moreover, since ST-GST is a mathematically designed framework, we are able to perform theoretical analysis and the related conclusions potentially shed some light on the design of spatio-temporal graph neural networks. The stability of ST-GST may improve the design of ST-GCNs, which are often vulnerable to small perturbations added to training data.
>
> To further improve the empirical performance of ST-GST on large-scale datasets, we also plan to introduce some trainable parameters into our current scattering architecture for future work. In this case we may be able to boost the classification accuracy, and find a trade-off between stability and empirical performance. One potential direction is related with this paper [Scattering GCN: Overcoming Oversmoothness in Graph Convolutional Networks (2020)].

---

### Official Review · AnonReviewer4 · 2020-10-28
**A very good extension of the graph scattering transform to the spatio-temporal case**

**Rating:** 9
**Confidence:** 2

**Review:**

1/ Summary of the paper

This paper proposes a novel spatio-temporal graph scattering transform (ST-GST) as an intermediate representation of time-varying signals on graphs, which can then be fed into a simple ML model. Building on previously introduced Graph Scattering Transform (GST), the authors investigate the technical question of how to use both the spatial and temporal dimensions, proposing either a joint construction (GST applied on product graph) or a separate one (keeping both dimensions separate conceptually, while mixing). Theoretical results valid in both cases guarantee stability to noise in the input signal or the spatial graph.
Numerical experiments show that the proposed method outperforms graph convolutional networks in the small data regime, and get a performance close to the state-of-the-art in the large data regime.

2/ Acceptance decision

Accept. This is a very good paper with novel contributions on the algorithmic side, theoretical guarantees and convincing numerical experiments.

3/ Supporting arguments

- In terms of novelty, this paper is the first to tackle the investigate the construction of a fixed representation of spatio-temporal graphs based on priors, using the graph scattering transform approach.
- In terms of technical contributions, the paper delves into the interesting technical question of how to build such a transform, a proposes convincing arguments to support its final approach, the separable ST-GST. Further, the authors provide theoretical guarantees of the stability of the ST-GST representation with respect to input and spatial graph variations, based on assumptions on the pre-defined filters used in the transform.
- Experimental results show that in the very low data regime (MSR Action3D dataset with only 288 training samples), the proposed approach improves upon the graph convolutional networks SOTA by ~5% of accuracy (87 vs 82.2). In the large data regime, the proposed method reaches a performance close to SOTA (73.1 vs 75.8% accuracy). The authors also show that the gap between GCN and ST-GST widens as the number of samples decreases.
- This paper is well motivated and well written. Although it is quite dense and with involved mathematical operations, it is quite easy to follow.

4/ Additional comments

- The authors do a very nice job of putting the joint and separable graph filters under similar notations (Eq. 1 and 2). When it comes to comparing both approaches at the end of Sec 3.2, I am not sure to understand correctly the statement that the separable approach is more « flexible » than the joint one. While I understand that separability brings an easier interpretability of filter responses in the spatial and temporal domain, it seems also that in terms of polynomials with atoms of the form $\lambda_s^p \otimes \Lambda_t^q$ for various $p$, $q$, the family spanned by separable transforms might be smaller than e.g. the strong product (although I’m not sure). Do you have any more insights in this direction?
- In the numerical results of Table 1, the separable approach gets the better results with much more scales in the temporal dim (20) than in the spatial one (5). In the joint approaches, a single number of scales (15) is used, and it spans both dimensions. Beyond the computational requirements of the joint approach, do you think this forced equality is an issue for the joint approach?
- In order to compare the significance of results in the low-data regime, standard deviations would have been appreciated, especially to understand the differences between the different joint approaches.
- Minor details: which graph shift is used in the experiments reported in the last page of the main paper? In Section 3.2, the authors state $S = A$ (for both spatial, temporal and joint cases), but appendix A defines $S$ as a lazy random walk for geometric graph wavelets and Appendix C states that geometric graph wavelets are used in the main paper. Similarly, which non linear activation is used eventually?

---

> ### Author Response · Authors · 2020-11-15
> **A few remarks on the comments raised by the Reviewer 4 (post 1)**
>
> [Response split into two posts: 1/2] We thank Reviewer 4 for her/his valuable feedback and suggestions for improving the manuscript. We addressed all the comments as described below.
>
> (Q1) "The family spanned by separable transforms might be smaller than e.g. the strong product..."
>
> The flexibility of separable graph filters means that we can design different graph filters ($\mathbf\{h\}$ and $\mathbf\{g\}$ in Eq.(1)) with independent filter length ($P$ and $Q$ in Eq.(1)) in the spatial and temporal domains. On the other hand, for joint graph filters, filter coefficients and length in the spatial and temporal domains are bonded together ($\mathbf\{h\}$ and $K$ in Eq.(2));  that is, we cannot change the filter in the spatial domain without changing the corresponding one in the temporal domain. This thus makes the separable graph filters more flexible than the joint graph filters.
>
> However, in terms of the representation power of these two formulations, the family spanned by joint graph filters (either Kronecker, Cartesian or strong product) and the family spanned by separable graph filters do not have a clear relationship that one is a subset of the other. We can show this through a toy example. Consider a graph filter with a strong product graph and the filter order $K=3$, we can write the kernel of strong product graph filter as $\mathbf\{H\}(\mathbf\{\Lambda\}_s\otimes \mathbf\{\Lambda\}_t+\mathbf\{\Lambda\}_s\otimes \mathbf\{I\}_T+\mathbf\{I\}_N\otimes \mathbf\{\Lambda\}_t)=\sum_\{k=0\}^2 h_k(\mathbf\{\Lambda\}_s\otimes \mathbf\{\Lambda\}_t+\mathbf\{\Lambda\}_s\otimes \mathbf\{I\}_T+\mathbf\{I\}_N\otimes \mathbf\{\Lambda\}_t)^k$. By expanding the expression and rearranging the coefficients, we can have the following coefficient matrix
> $$
> C=\begin\{bmatrix\}
>         h_0 & h_1 & h_2 \\\\
>         h_1 & h_1+2h_2 & 2h_2 \\\\
>         h_2 & 2h_2 & h_2
>     \end\{bmatrix\},
> $$
> where $C_\{ij\}$ means the coefficient of term $\mathbf\{\Lambda\}_s^\{i-1\}\otimes \mathbf\{\Lambda\}_t^\{j-1\}$. On other hand, the kernel of separable graph filter with $P=Q=3$ can be written as $\mathbf\{A\}(\mathbf\{\Lambda\}_s)\otimes \mathbf\{B\}(\mathbf\{\Lambda\}_t)=(\sum_\{k=0\}^2 a_k \mathbf\{\Lambda\}_s^k)\otimes (\sum_\{k=0\}^2 b_k \mathbf\{\Lambda\}_t^k)$. By the same rearranging procedure, we can have the coefficient matrix for separable graph filter as
> $$
> D=\begin\{bmatrix\}
>         a_0b_0 & a_0b_1 & a_0b_2 \\\\
>         a_1b_0 & a_1b_1 & a_1b_2 \\\\
>         a_2b_0 & a_2b_1 & a_2b_2
>     \end\{bmatrix\} = \begin\{bmatrix\}
> a_0 \\\\
> a_1 \\\\
> a_2
> \end\{bmatrix\} \begin\{bmatrix\}
> b_0 & b_1 & b_2
> \end\{bmatrix\}.
> $$
> If the family spanned by one method is a subset of the other, then one of the coefficient matrix must be a special case of the other one. On one hand, it is obvious that $D$ is always a rank 1 matrix, while $C$ could have rank 1, 2, or 3. So $C$ is not a special case of $D$. On the other hand, $C$ is always a symmetric matrix, but $D$ can be either symmetric or non-symmetric. Furthermore, the ratio between several pairs of elements in $C$ is fixed, for example $C_\{32\}/C_\{31\}$ is always 2 in this case; however, $D_\{32\}/D_\{31\}=b_1/b_0$ can be an arbitrary value. So $D$ is also not a special case of $C$. Therefore, the families spanned by two different methods do not have any simple relationship that one is a subset of the other. This conclusion also holds for the Kronecker and Cartesian products.
>
> In short, the joint and separable graph filters are two different design methods for spatio-temporal graphs. Though the representation power of separable graph filters is not necessarily much stronger than joint ones, separable design enjoys the flexibility, computation efficiency and straightforward interpretation.

---

> ### Author Response · Authors · 2020-11-16
> **A few remarks on the comments raised by the Reviewer 4 (post 2)**
>
> [Response split into two posts: 2/2]
>
> (Q2) “do you think this forced equality is an issue for the joint approach”
>
> We thank Reviewer 4 for this insightful question, and we will include the discussion here in our revision. The number of scales is essentially the number of graph filters used at each layer. For most spatio-temporal graph signals, the number of time stamps is much larger than the number of spatial locations such as joints. Thus it is intuitive to use more scales/filters in the temporal domain than in the spatial domain. Such a claim is also supported by the experimental results shown in Table 1, where separable ST-GST (5, 20, 3) performs better than separable ST-GST (5, 10, 3).
>
> As for joint ST-GST, at each scale, the joint graph filter forces the spatial and temporal domains to share the same set of filter coefficients and length. Therefore, such kind of “``forced equality” with respect to spatial and temporal domains exists not only for the number of scales, but also for the filter coefficients and filter length at each scale. All these limitations can restrict the performance of the joint ST-GST and make it not comparable to the performance of the separable ST-GST. Moreover, we believe that the fundamental limitation of joint ST-GST is the shared filter coefficients and length, as the product graph contains both spatial and temporal connections, and so we cannot adapt the filter coefficients and length separately along space and time (see also response to (Q1)). We have run a toy experiment to support our claim. Note that 20 independent trails are run for each setting to obtain the mean and standard deviation of classification accuracy. For MSR Action3D dataset, with the other environment setting the same as stated in our manuscript, separable ST-GST (5, 5, 3) can achieve the accuracy of $73.4\\%\\pm 0.8\\%$, while joint-Kronecker-product ST-GST (5, 3) with $46.3\\%\\pm 1.2\\%$, joint-Cartesian-product ST-GST (5, 3) with $42.2\\%\\pm 1.1\\%$ and joint-strong-product ST-GST (5, 3) with $45.0\\%\\pm 1.2\\%$. We can see that even if the number of scales/filters is the same for the spatial and temporal domains, separable ST-GST still outperforms all joint ST-GST methods by a huge margin.
>
> (Q3) "Add standard deviations in low-data regime to understand the differences between the different joint approaches"
>
> We thank Reviewer 4 for this valuable suggestion. Due to the time limit of discussion phase we are not able to run all our experiments multiple times now, but we have rerun part of our experiments on MSR Action3D dataset, especially for joint approaches, to obtain the standard deviations of classification accuracy. The new results with standard deviations are listed here. Note that 20 independent trails are run for each setting here.
>
> * Separable ST-GST (5, 5, 3): $73.4\\%\\pm 0.8\\%$
> * Separable ST-GST (5, 20, 3): $86.7\\%\\pm 0.4\\%$
> * Joint-Kronecker-product ST-GST (5, 3): $46.3\\%\\pm 1.2\\%$
> * Joint-Cartesian-product ST-GST (5, 3): $42.2\\%\\pm 1.1\\%$
> * Joint-strong-product ST-GST (5, 3): $45.0\\%\\pm 1.2\\%$
> * Joint-Kronecker-product ST-GST (15, 3): $59.6\\%\\pm 0.5\\%$
> * Joint-Cartesian-product ST-GST (15, 3): $58.6\\%\\pm 1.0\\%$
> * Joint-strong-product ST-GST (15, 3): $60.0\\%\\pm 1.0\\%$
>
> We can see that the standard deviations are comparable in all these methods, and therefore the conclusion that separable ST-GST consistently outperforms joint ST-GST still holds. If time permits, we will run more experiments and update these results in revised version of this paper.
>
> (Q4) "Detailed experiment setting"
>
> In our experiments, we use the lazy random walk matrix as the graph shift, which is also adopted in geometric graph wavelets. In Section 3.2, we consider graph adjacency matrix (broadly used in graph signal processing literature) for simplicity in notation, because the corresponding eigen-decompositions for three kinds of product graphs have the simple, clear and compact forms (see reference [Big Data Analysis with Signal Processing on Graphs (2014)]). Although similar conclusion can be drawn for the lazy random walk matrix by following the same procedure, the results are much more involved and hard to interpret. Note that other valid and commonly used choices for graph shift are graph Laplacian matrix and its normalized version. We will update the paper to reflect this.
> As for the choice of nonlinear activation, we use absolute value function. The main reason is that absolute value function is energy-preserving. This follows the same setting with [Invariant Scattering Convolution Networks (2013)] and [Stability of graph scattering transforms (2019)].

---

### Official Review · AnonReviewer1 · 2020-10-29

**Rating:** 6
**Confidence:** 4

**Review:**

I will add it tomorrow

---

> ### Author Response · Authors · 2020-11-15
> **A response to Reviewer 1**
>
> We thank Reviewer 1 for her/his review of the manuscript. We are also happy to answer any other additional questions or comments that Reviewer 1 may have.

---

### Author Response · Authors · 2020-11-24
**Revision of our manuscript**

We again thank all reviewers for their constructive suggestions on improving our manuscript. We have uploaded a revision of the manuscript. Detailed changes in revision are:
* More discussion about the comparison between separable and joint graph filters at the end of Section 3.2
* Motivation of using spatio-temporal graph wavelets at the begining of Section 3.3
* More details about the experimental setup
* More discussion about the experimental results
* More experiments to get the standard deviations of classification accuracy in appendix
* Correct some typos

---

### Decision · Program_Chairs · 2021-01-07
**Final Decision**

**Decision:**

Accept (Poster)

**Comment:**

This paper studies extensions of the Scattering Graph Transform to spatio-temporal domains. By exploring several design choices for spatio-temporal wavelet filters, the authors provide a solid and broad study of such predefined represenatations, including stability analysis as well as extensive empirical evaluations.
Reviewers were generally favorable, and highlighted the importance of this method as providing a simple yet powerful baseline for spatio-temporal graph prediction tasks that requires no training. Despite some concerns about lack of analysis of the empirical results, the AC believes this work will provide a valuable baseline for future research and therefore recommends acceptance as a poster.